# CLUE: Fine-Grained Self-Supervised Learning with Multi-Level Regularization

## Abstract

Self-supervised learning (SSL) has achieved strong results on coarse-grained tasks but often struggles with fine-grained recognition, where categories differ only by subtle local cues. For strong downstream transfer, features must form compact within-class clusters with large inter-class margins at the fine level. However, standard SSL losses either over-separate visually similar subcategories by treating all non-positives as equally negative, or overlook part-based evidence and thus merge them under coarse prototypes. We propose a multi-level regularization framework that improves clustering across granularities. At the *global level*, a soft variant of InfoNCE reduces false negatives and enhances class separation; at the *part level*, clustering on local descriptors preserves subtle intra-class distinctions, at the *instance level*, semantic descriptions from vision–language models provide attribute-level anchors. Together, these components yield representations with balanced clustering across granularities. Experiments on multiple fine-grained datasets show consistent improvements in both classification and retrieval, validating the effectiveness of our approach for fine-grained SSL.

## 1 Introduction

Self-supervised learning (SSL) has achieved remarkable success in learning visual representations without human annotations, enabling models to exploit large-scale unlabeled data. Recent advances—spanning contrastive methods (Chen et al., 2020a; He et al., 2020) and non-contrastive paradigms (Grill et al., 2020; Bardes et al., 2021; Oquab et al., 2023; Siméoni et al., 2025) have delivered strong performance on downstream tasks including image classification, object detection, and semantic segmentation. However, SSL still underperforms in fine-grained visual recognition (FGVR), where the objective is to distinguish visually similar subcategories (e.g., bird species or car models). Such tasks impose stricter requirements on the discriminative ability of learned representations, as subtle local differences must be captured reliably.

Recent studies have revealed that SSL representations tend to exhibit clustering behavior, where learned features are naturally grouped into semantic categories. By decomposing SSL objectives into an invariance term and a regularization term, (Ben-Shaul et al., 2023) demonstrate that while the invariance term saturates early (e.g., in VICReg (Bardes et al., 2021)), the regularization term continues to shape the geometry of feature space and is primarily responsible for the emergence of semantic clustering. Such clustering structures yield well-formed feature arrangements that are beneficial for downstream transfer, making the design of effective regularization particularly critical.

Although a number of SSL methods have been explored for fine-grained visual recognition (FGVR), their performance remains limited. FGVR demands discrimination between visually similar subcategories (Shu et al., 2023; Wang et al., 2024), which requires preserving subtle and localized cues throughout pre-training. We argue that the bottleneck lies less in invariance learning and more in *granularity mismatch* in regularization: current objectives shape feature geometry mainly at a coarse level, but lack explicit guidance at fine levels. As a consequence (Fig. 1), two failure modes frequently arise: **Over-dispersion (left).** Fine categories scatter excessively around coarse clusters: intra-class coherence is weak at the fine level despite clear coarse-level separation. **Over-collapse (middle).** Fine categories collapse toward their coarse centers: fine-level distinctions vanish even though coarse clusters are well formed. **Ideal structure (right).** Coarse categories are well separated while fine-category centers remain distinct with balanced spacing. In Sec. 3, these phenomena are

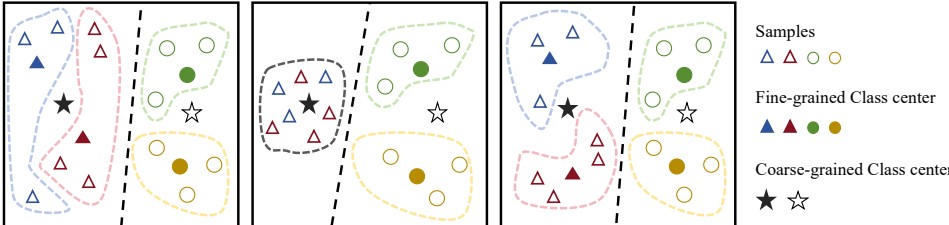

Figure 1: Illustration of granularity-related failure modes in self-supervised learning. **Left:** Over-dispersion, where coarse categories are separated but fine-grained categories scatter too widely. **Middle:** Over-collapse, where fine-grained categories collapse into coarse centers, losing intra-coarse distinctions. **Right:** Ideal structure, where coarse categories remain separated and fine-grained categories form distinct, balanced clusters. In this visualization, each *shape* represents a coarse category, each *color* within a shape denotes a fine category, solid symbols mark fine-category centers, and stars mark coarse-category centers.

quantified by geometry-aware metrics (e.g., coherence and dispersion at both coarse and fine levels) and empirical results show that multi-granularity regularization *enhances the retention of fine-grained cues*, which correlates with improved recognition performance.

Motivated by these observations, and building on the analysis of (Ben-Shaul et al., 2023) which links well-structured clustering with strong downstream transfer, we design **CLUE**, a framework that explicitly strengthens fine-grained capabilities through **CLU**st**E**ring-aware regularization. Concretely, our approach integrates three complementary components: **(1) Class-level regularization.** We employ a soft contrastive loss that addresses the limitations of InfoNCE (Weng et al., 2025). Rather than treating all non-positives as equally negative, the soft assignment reweights pairwise relations, reducing false negatives and yielding more coherent class-level structures. **(2) Part-level regularization.** To capture subtle local differences, we extend the clustering objective to part-aware representations derived from intermediate feature maps. Applying soft assignment at this level encourages the model to disentangle fine-grained subcategories within the same coarse class, mitigating over-collapse. **(3) Instance-level regularization.** Finally, we incorporate textual guidance from vision–language models (VLMs). By aligning image features with automatically generated textual embeddings, the model is anchored to diverse semantic directions, which helps instance-level collapse and further enhances fine-grained discrimination. Together, these three components form a multi-level regularization framework that balances coarse- and fine-grained structures, enabling SSL models to achieve consistently stronger performance on fine-grained recognition benchmarks.

We evaluate our approach on widely used fine-grained benchmarks, including CIFAR-100 (Krizhevsky et al., 2009), Stanford Cars (Krause et al., 2013), CUB-200 (Wah et al., 2011), and FGVC Aircraft (Maji et al., 2013). Across these datasets, our method consistently surpasses strong SSL baselines, yielding notable gains in top-1 accuracy and retrieval performance. In addition, we demonstrate that the proposed components help alleviate the granularity-related failure modes discussed above.

## 2 RELATED WORK

### 2.1 FINE-GRAINED SELF-SUPERVISED LEARNING

Self-supervised learning (SSL) has achieved remarkable progress in visual representation learning. Contrastive methods such as SimCLR (Chen et al., 2020a) and MoCo (He et al., 2020) learn by enforcing consistency between augmented views of the same image while repelling negatives, effectively shaping instance-discriminative features. Later approaches including Barlow Twins (Zbontar et al., 2021) and VICReg (Bardes et al., 2021) reduce redundancy and decorrelate features without explicit negatives, while reconstruction-based paradigms such as MAE (He et al., 2022) and BEiT (Bao et al., 2021) leverage masked image modeling to capture global structures. These methods yield strong general-purpose representations and have become standard baselines in SSL.

However, applying these frameworks directly to fine-grained recognition reveals critical limitations. Instance-level objectives primarily emphasize global alignment and overlook subtle local cues, which are essential for distinguishing visually similar subcategories (Cole et al., 2022). Contrastive methods can suffer from over-dispersion by aggressively separating semantically related samples, while redundancy-reduction or reconstruction-based methods may underexploit discriminative part-level information, leading to over-collapse within coarse categories. Consequently, although SSL methods transfer well to generic tasks, their ability to capture fine-grained semantics remains limited. Our work addresses this gap by introducing explicit regularization across multiple granularities to preserve both coarse- and fine-level structures.

## 2.2 Hierarchical and part-aware representation learning

A number of self-supervised methods for FGVR explicitly model hierarchy or parts. Prototype-based approaches (Tan et al., 2025) construct semantic prototypes and refine them stage-wise, while Particle (Saha & Maji, 2023) discovers object parts and applies contrastive learning over the discovered regions. HIRL (Xu et al., 2022) learns hierarchical image representations and S-JEA (Manová et al., 2023) stacks joint-embedding branches to capture multi-scale invariances; meanwhile, ViT-based SSL methods such as DINOv2 and CMD (Oquab et al., 2023; Bi et al., 2025) show that multi-/local-crop strategies can implicitly enhance fine-grained modeling. Overall, these works tend to operationalize hierarchy through architectural components (e.g., prototype heads, part branches, stacked embeddings) or carefully designed cropping schemes.

CLUE is related to these methods in that it also incorporates a lightweight part-extraction module, but its focus is different. Rather than designing a specific hierarchical architecture, CLUE starts from a *granularity-aware clustering* perspective and treats global features, part descriptors, and attribute-level (text) anchors as three coupled views that jointly shape the regularization term of the SSL objective. The part module in CLUE is used as a vehicle to extend a soft clustering loss to the local level, with the same assignment structure shared across global and part features, while the VLM-guided term introduces additional attribute-level anchors. This shifts the emphasis from building an explicit hierarchy of predictors to explicitly controlling how clusters form and separate across granularities, and makes CLUE complementary to prior hierarchical SSL approaches (Tan et al., 2025; Saha & Maji, 2023; Xu et al., 2022; Manová et al., 2023).

## 2.3 Vision-Language Models and Semantic Guidance

Large-scale vision–language models (VLMs) such as CLIP (Radford et al., 2021) and ALIGN (Jia et al., 2021) have demonstrated remarkable transferability by aligning images and text in a joint embedding space through large-scale contrastive pretraining on image–text pairs. These models enable zero-shot and open-vocabulary recognition and have inspired follow-up work (Mu et al., 2022; Gu et al., 2021) that integrates linguistic cues into visual representation learning. However, prior efforts have primarily employed VLMs for global or class-level supervision, while their potential to enhance fine-grained, part-level representations remains underexplored. In this work, we exploit VLM-generated semantic cues as external anchors to guide self-supervised learning, thereby improving fine-grained recognition especially under limited annotation.

## 3 Granularity-Aware Collapse in SSL

### 3.1 Preliminaries

**CDNV (Class-Distance Normalized Variance).** Following Galanti et al. (2021), Class-Distance Normalized Variance (CDNV) quantifies *feature-variability collapse*, i.e., the extent to which samples from the same class are compressed into a narrow region of the feature space. Let $f : \mathbb{R}^d \to \mathbb{R}^p$ denote the representation function, and let $S_1, \dots, S_C \subset \mathbb{R}^d$ be disjoint sets of samples belonging to different classes. For any pair of classes $(S_i, S_j)$, CDNV is defined as

$$V_f(S_i, S_j) = \frac{\mathrm{Var}_f(S_i) + \mathrm{Var}_f(S_j)}{2 \left\| \mu_f(S_i) - \mu_f(S_j) \right\|_2^2},$$
(1)

where $\mu_f(S)$ is the class centroid in the feature space and $\text{Var}_f(S) = \mathbb{E}_{x \in S}\|f(x) - \mu_f(S)\|_2^2$ is the within-class variance. The overall CDNV is obtained by averaging over all class pairs:

$$\text{CDNV} = \underset{i \neq j}{\text{Avg}}\, V_f(S_i, S_j). \tag{2}$$

Intuitively, CDNV compares intra-class variance with inter-class centroid distance. A lower CDNV implies that samples within a class form tight, compact clusters that are still well separated from other classes. Hence, CDNV is a useful diagnostic for whether learned features balance compactness and separability across categories.

**NCC (Nearest Class-Center Separability).** Normalized Class Confusion (NCC) evaluates how well class centroids can serve as decision boundaries in the learned feature space. Formally, the nearest class-center (NCC) classifier is defined as

$$h(x) = \arg \min_{c \in [C]}\, \big\|f(x) - \mu_f(S_c)\big\|_2, \tag{3}$$

where $\mu_f(S_c)$ denotes the centroid of class $c$ in the feature space. The NCC accuracy is then computed by applying $h(\cdot)$ to all samples. High NCC accuracy implies that embeddings are naturally organized around their class centroids, i.e., samples lie close to the correct centroid and far from others. This centroid-like geometry is a hallmark of SSL representations and provides an interpretable measure of class separability.

**Granularity Variants.** To capture hierarchical structure, we extend CDNV and NCC to multiple granularities, yielding richer insights than standard fine-level metrics. Consider a dataset with $C_{\text{fine}}$ fine classes grouped into $C_{\text{coarse}}$ coarse classes via a mapping $\pi : [C_{\text{fine}}] \to [C_{\text{coarse}}]$. Let $S_i$ denote the sample set of fine class $i$, and $S_A = \bigcup_{i:\pi(i)=A} S_i$ the union of fine classes belonging to coarse class $A$. We define three variants:

(1) $\text{CDNV}_A$. For each coarse class $A$, we compute CDNV over the fine classes $\{S_i : \pi(i) = A\}$. This reflects the compactness and separability of fine categories within $A$. A lower $\text{CDNV}_A$ means that fine classes in the same coarse group are tightly clustered around their centroids yet remain distinguishable. In contrast to the global $\text{CDNV}_{\text{all}}$, which aggregates over all fine classes, $\text{CDNV}_A$ focuses specifically on intra-coarse compactness and separability.

(2) $\text{NCC}_{\text{fine}}$. This metric computes NCC accuracy using fine-class labels and centroids $\mu_f(S_i)$ across all samples. Higher accuracy indicates that embeddings align closely with their fine-class centroids, demonstrating stronger fine-level discriminability.

(3) $\text{NCC}_{\text{coarse}}$. This variant computes NCC accuracy using coarse-class labels and centroids $\mu_f(S_A)$. A higher score suggests that fine-class samples are well organized around their coarse-class centroids, showing that the coarse semantic structure is well preserved in the representation space.

In summary, $\text{CDNV}_A$ captures intra-coarse compactness and separability among fine classes within each coarse group, whereas $\text{CDNV}_{\text{all}}$ reflects global relations across all fine classes. Similarly, $\text{NCC}_{\text{fine}}$ and $\text{NCC}_{\text{coarse}}$ evaluate alignment with fine- and coarse-level centroids, respectively. Taken together, these metrics provide a complementary suite for quantifying clustering quality across semantic granularities.

## 3.2 Observation

CIFAR-100 provides a hierarchical structure with 100 fine classes organized into 20 coarse categories (five per coarse group). We pretrain a ResNet-50 on this dataset and evaluate the learned representations using the granularity-aware metrics introduced in Sec. 3.1. As shown in Fig. 2, the three geometric patterns illustrated in Fig. 1, over-dispersion, over-collapse, and the ideal structure, indeed manifest empirically. For reference, the dataset-level mean $\text{NCC}_{\text{fine}}$ on the training split is $59.16\%$; unless otherwise specified, all reported results are computed on the training set.

**Over-dispersion (e.g., coarse ID 14).** Here $\text{NCC}_{\text{coarse}}$ is clearly above the dataset mean, yet $\text{CDNV}_A$ for its five fine classes is markedly elevated and $\text{NCC}_{\text{fine}}$ (34.28%) is depressed. This indicates that fine classes scatter widely despite good coarse-level separation, matching the left pattern in Fig. 1.

**Over-collapse (e.g., coarse ID 17).** In this case $\text{CDNV}_A$ is notably low and $\text{NCC}_{\text{coarse}}$ remains high, but $\text{NCC}_{\text{fine}}$ (39.04%) stays below the mean. Fine categories therefore contract toward the coarse centroid and lose intra-coarse distinctions, corresponding to the middle pattern in Fig. 1.

**Ideal structure (e.g., coarse ID 9).** Here both $\text{NCC}_{\text{fine}}$ (73.52%) and $\text{NCC}_{\text{coarse}}$ are high, while $\text{CDNV}_A$ remains relatively low. Fine clusters are compact and separable, nested within well-separated coarse groups, consistent with the right pattern in Fig. 1.

These observations reveal that standard SSL can break down at the fine level in

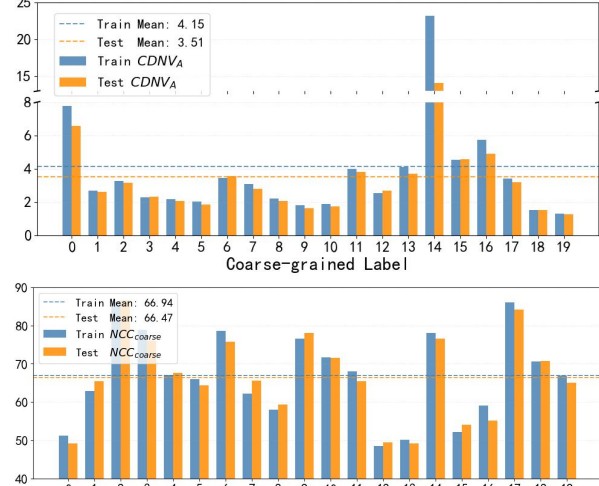

Figure 2: Empirical observation of granularity-aware metrics on CIFAR-100. **Top:** $\text{CDNV}_A$ across training epochs, where lower values indicate tighter and more distinguishable fine-class clusters within each coarse group. **Bottom:** $\text{NCC}_{\text{coarse}}$, where higher accuracy reflects stronger alignment of samples with their coarse-class centroids, i.e., clearer coarse-level separation.

two opposite ways: *over-dispersion* or *over-collapse*, even when coarse structure appears satisfactory. The root cause lies in how current objectives regulate geometry. On the one hand, contrastive losses typically treat all non-positives as equally negative, which pushes apart visually similar (but semantically related) samples and leads to over-dispersion. On the other hand, existing methods underexploit *local* image cues that encode subtle distinctions, so fine classes may collapse toward their coarse centroid. Without labels, the definition of "equivalence" remains ambiguous, making this trade-off inherently challenging. Motivated by this analysis, we seek to preserve fine-grained structure while maintaining coarse-level separation. In Sec. 4, we present a multi-level regularization framework that integrates: (i) a *global-level* soft-alignment term to temper the effect of hard negatives, (ii) a *part-level* term that leverages local descriptors to preserve intra-coarse distinctions, and (iii) *semantic* cues from vision–language models to anchor instance-level uniqueness.

## 4 METHOD

### 4.1 SOFT-INFONCE AS A STRONGER BASELINE REGULARIZATION

Following (Ben-Shaul et al., 2023), the SSL objective can be decomposed into an invariance term and a regularization term. While the invariance term quickly saturates in early training, the regularization term continues to decrease and plays the dominant role in shaping semantic clustering. An effective regularization should not only prevent representational collapse but also promote meaningful class separation. However, the standard InfoNCE loss treats all non-positive pairs as equally negative, ignoring the underlying fine-grained relations. This indiscriminate repulsion often leads to either over-dispersion or collapse toward coarse-level centers.

To mitigate this limitation, we adopt ReSA (Weng et al., 2025) as our baseline. ReSA replaces the hard one-to-one target distribution in InfoNCE with a soft assignment matrix, thereby alleviating the uniform repulsion among non-positive samples. Formally, given a batch of $2m$ augmented samples, let $Z, Z' \in \mathbb{R}^{d \times m}$ denote the embeddings, and define the similarity matrix $S_Z = Z^\top Z'$. Standard InfoNCE minimizes the cross-entropy loss with the identity matrix $I$ as the target distribution:

$$L_{\text{InfoNCE}} = -\frac{1}{2m} \sum_{i,j} I_{ij} \log D(S_Z)_{ij} + I_{ji} \log D(S_Z^\top)_{ji}, \tag{4}$$

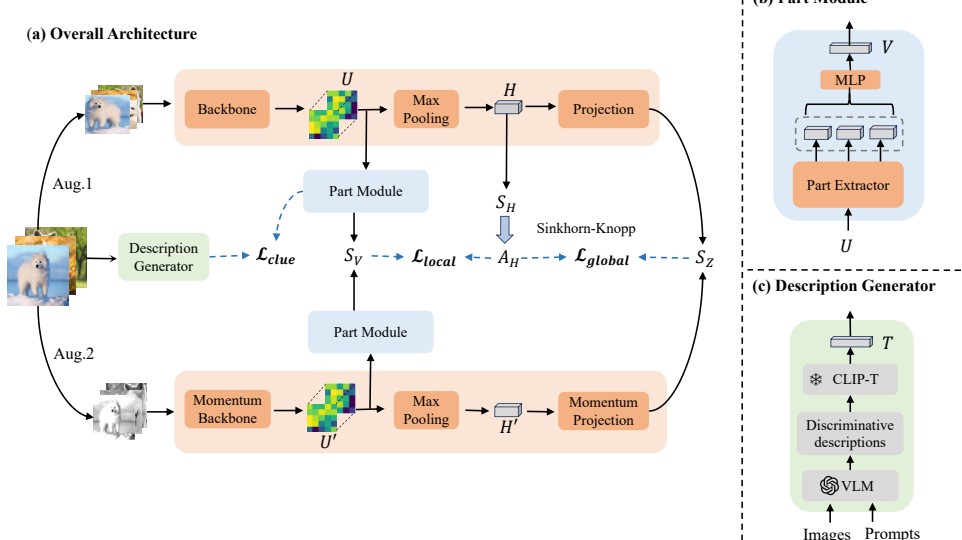

Figure 3: Pipeline of the proposed clustering-guided self-supervised learning framework (**CLUE**). (a) Overall architecture: the Sinkhorn–Knopp algorithm is applied to the similarity matrix $S_H$ to produce soft assignments $A_H$, which guide both global and local contrastive losses. (b) Part-assignment module: semantic part features are extracted from activation maps and used to capture fine-grained distinctions. (c) Description generator: a vision–language model (VLM) produces discriminative textual descriptions, which are embedded by CLIP-T into the text feature space to provide semantic guidance for representation learning.

where $D(\cdot)$ denotes the row-wise softmax normalization with temperature $\tau$. ReSA introduces an online clustering step that produces a doubly-stochastic assignment matrix $A \in \mathbb{R}^{m \times m}$ via the Sinkhorn–Knopp algorithm. The resulting soft target distribution leads to the following soft contrastive loss:

$$L_{\text{soft-InfoNCE}} = -\frac{1}{2m} \sum_{i,j} A_{ij} \log D(S_Z)_{ij} + A_{ji} \log D(S_Z^\top)_{ji}. \tag{5}$$

When $A = I$, this reduces to standard InfoNCE. The doubly-stochastic constraint enforces balanced assignments and prevents trivial collapse to a single mode. In practice, $A$ is computed from the encoder's intermediate features (rather than the projection head), which have been shown to provide more stable clustering (Weng et al., 2025).

Compared with InfoNCE, the gradients of $L_{\text{soft-InfoNCE}}$ reweight pairwise relations: similar samples receive larger positive weights, which reduces false negatives and preserves fine-grained structures, while truly dissimilar samples remain repelled. Thus, soft-InfoNCE offers a stronger and more principled form of semantic regularization, and we use it as the foundation for our multi-level extension in Sec. 4.

### 4.2 EXTENDING THE CLUSTERING LOSS TO PART-LEVEL

While the soft-InfoNCE in Sec. 4.1 mitigates over-dispersion at the *global* level, relying only on a holistic feature vector $h$ is insufficient for distinguishing instances within the same coarse class. To explicitly capture fine-grained cues, we extend the clustering loss to operate on *part-level* features.

**Part-aware representation.** Given a convolutional feature map $F \in \mathbb{R}^{C \times H \times W}$, we adopt a VLAD-like residual aggregation (Arandjelovic et al., 2016) to obtain $K$ part descriptors: $P = \{p_1, p_2, \ldots, p_K\}$, $p_k = \sum_u \alpha_{uk} (f_u - c_k)$, where $f_u$ is the local descriptor at spatial position $u$, $c_k$ is the $k$-th part centroid, and $\alpha_{uk}$ is the normalized assignment weight across parts. The resulting descriptors are flattened and concatenated into a part-aware vector $v = [p_1, p_2, \ldots, p_K]$.

**Part-level soft contrastive loss.** We replace the similarity matrix in Eq. equation 4 with part-aware similarities $S_V = V^\top V'$, where $V$ stacks part-aware vectors from a batch. The same soft assignment matrix $A$ is reused to construct the part-level loss:

$$L_{\text{local}} = -\frac{1}{2m} \sum_{i,j} A_{ij} \log D(S_V)_{ij} + A_{ji} \log D(S_V^\top)_{ji}. \tag{6}$$

This formulation mirrors Eq. equation 5, but in the part-aware coordinate system. Rather than acting as an auxiliary objective, it imposes stronger regularization directly on discriminative local subspaces.

**Geometric effect.** VLAD-style residuals encourage local descriptors to cluster around multiple part centroids, naturally forming multi-modal fine-grained subclusters within each coarse class. The gradient of $L_{\text{part}}$, proportional to $D(S_V) - A$, enforces consistency between these subclusters and the global soft assignment $A$. As a result, fine-grained categories are pushed apart while maintaining intra-coarse coherence, thereby mitigating over-collapse and preserving subtle distinctions that would otherwise be absorbed by coarse-class centers.

### 4.3 VLM-Driven Guidance for Instance Information

Vision–language models (VLMs) excel at broad recognition and reasoning but often lack the granularity needed for fine-grained categorization (Peng et al., 2024; Jing et al., 2024). Even when a VLM confuses closely related species at the *class* level, it typically produces *reliable region-level descriptions* (e.g., attributes, textures, part shapes) that domain experts rely on to separate fine categories (Zhao et al., 2025). Thus, although class predictions may be unreliable, the accompanying textual cues provide *stable fine-grained semantics* that can serve as anchors for representation learning. Building on this observation (Fig. 4), we regard a pre-trained VLM as an external expert and incorporate its descriptions as semantic priors, following the spirit of recent VLM-assisted approaches (Bang et al., 2024; El Banani et al., 2023; Shrivastava et al., 2021).

Complementary to the global and part-level objectives, text guidance addresses both over-dispersion and over-collapse. Images sharing fine-grained attributes are drawn toward the same textual anchors, which reduces variance along attribute dimensions and lowers CDNV without collapsing to zero. Meanwhile, within a common coarse category, samples with distinct attribute signatures are attracted to different anchors, effectively partitioning a coarse cluster into multiple fine-level centers.

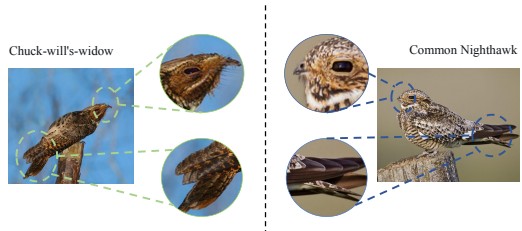

Figure 4: Comparison between *Chuck-will's-widow* and *Common Nighthawk*. Although a VLM may confuse them at the class level, it correctly highlights fine-grained cues such as the "small, up-tilted bill fringed with whisker-like bristles" and the "rusty-orange bars on the tail and wing edges," which serve as key attributes for distinguishing the two genera.

To generate such anchors, we design a simple prompting strategy that *(a)* produces a global summary of the image, *(b)* enumerates salient parts and attributes, and *(c)* condenses these into a concise description. After light de-duplication, each description is encoded by a frozen CLIP text encoder to yield $t_i$. This semantic prior stabilizes early clustering and injects fine-grained supervision without labels. Given an image embedding $v_i$ and its corresponding text embedding $t_i$, we align them using a temperature-scaled cross-entropy:

$$\mathcal{L}_{\text{text}} = -\frac{1}{m} \sum_{i=1}^{m} \log \frac{\exp(\langle v_i, t_i \rangle / \tau_t)}{\sum_{j=1}^{m} \exp(\langle v_i, t_j \rangle / \tau_t)}. \tag{7}$$

Without loss of generality, we combine the three objectives into a single loss:

$$\mathcal{L}_{\text{total}} = \alpha\,\mathcal{L}_{\text{global}} + \beta\,\mathcal{L}_{\text{local}} + \gamma\,\mathcal{L}_{\text{text}}, \tag{8}$$

where $\mathcal{L}_{\text{global}}$ is the soft-InfoNCE loss from Sec. 4.1, $\mathcal{L}_{\text{local}}$ is the part-aware loss from Sec. 4.2, and $\mathcal{L}_{\text{text}}$ is the VLM-guided alignment from Sec. 4.3. The coefficients $\alpha, \beta, \gamma$ balance the relative strength of each term; unless otherwise specified, we set them all to 1.

Table 1: Fine-grained classification accuracy (%) and retrieval performance (Rank-1 / Rank-5, %) on CUB200, Stanford Cars, and FGVC-Aircraft. Use a ResNet-50 backbone unless otherwise specified.

| Method | Classification | | | Retrieval | | | | | |
| | CUB200 | Cars | Aircraft | CUB200 | | Cars | | Aircraft | |
| | | | | Rank1 | Rank5 | Rank1 | Rank5 | Rank1 | Rank5 |
|---|---|---|---|---|---|---|---|---|---|
| SimSiam (Chen & He, 2021) | 46.75 | 45.72 | 38.52 | 16.24 | – | 12.45 | – | 18.49 | – |
| MoCo v2 (Chen et al., 2020b) | 63.98 | 62.02 | 51.13 | 39.72 | 67.14 | 30.51 | 56.15 | 30.02 | 52.87 |
| LEWEL (Huang et al., 2022) | 64.59 | 62.91 | 51.90 | 39.91 | – | 32.36 | – | 31.09 | – |
| Contrastive Crop (Peng et al., 2022) | 64.23 | 63.29 | 52.04 | 39.84 | – | 32.71 | – | 30.37 | – |
| SAM-SSL-Bilinear (Shu et al., 2022) | 64.94 | 62.85 | 52.83 | 40.08 | – | 33.19 | – | 30.52 | – |
| MAE (He et al., 2022) | 38.92 | 43.30 | 55.72 | 12.45 | 28.49 | 13.04 | 27.97 | 31.75 | 57.72 |
| BEiT (Bao et al., 2021) | 25.70 | 36.48 | 46.62 | 5.62 | 15.03 | 11.45 | 25.87 | 24.49 | 31.02 |
| Barlow Twins (Zbontar et al., 2021) | 33.45 | 31.91 | 34.77 | 15.24 | 38.35 | 11.99 | 30.17 | 16.32 | 35.55 |
| VICReg (Bardes et al., 2021) | 37.78 | 30.80 | 36.00 | 17.02 | 42.85 | 12.31 | 30.52 | 14.43 | 37.02 |
| LCR (Shu et al., 2023) | 65.24 | 63.96 | 53.22 | 41.26 | – | 34.74 | – | 31.55 | – |
| LDF (Wang et al., 2024) | 66.17 | 65.60 | 55.28 | 42.06 | 69.59 | 35.81 | 61.94 | 33.27 | 56.80 |
| PAPN (Tan et al., 2025) | 69.93 | 67.48 | **60.13** | 45.39 | 72.81 | 35.98 | 59.94 | 35.13 | 58.75 |
| ReSA (Weng et al., 2025) | 65.82 | 64.76 | 56.70 | 42.53 | 71.31 | 34.92 | 60.46 | 34.64 | 58.84 |
| **CLUE (Ours)** | **69.62** | **72.66** | 58.59 | **48.53** | **74.71** | **43.45** | **69.49** | **40.66** | **63.56** |
| EsViT (Swin-T) (Li et al., 2021) | 70.54 | 59.12 | 55.18 | 43.48 | 73.08 | 31.95 | 58.40 | 27.06 | 53.02 |
| LoDisc (ViT-B) (Shi et al., 2025) | 73.23 | 69.72 | 62.17 | 45.89 | 72.75 | 41.55 | 67.24 | 41.49 | 68.59 |
| CLUE (ViT-B) (Ours) | 77.83 | 75.67 | 62.76 | 61.93 | 83.91 | 52.05 | 78.14 | 40.26 | 65.11 |

# 5 EXPERIMENTS

In this section, we evaluate the performance of the proposed method on three fine-grained image datasets: Caltech UCSD-Birds (CUB200) (Wah et al., 2011), Stanford Cars (Cars) (Krause et al., 2013), FGVC-Aircraft (Aircraft) (Maji et al., 2013).

## 5.1 SETTINGS

### 5.1.1 IMPLEMENTATION DETAILS

For all experiments, we adopt ResNet-50 (He et al., 2016) as the backbone and a standard three-layer MLP projector. Following common practice (He et al., 2020; Grill et al., 2020; Caron et al., 2020), we use a momentum encoder with coefficient $0.999$. The number of part centroids in the part-level module is set to $K = 3$, which provides a good balance between capturing discriminative parts and computational efficiency. All models are optimized with SGD under a cosine learning-rate schedule. Our training protocol is aligned with prior work (Shu et al., 2023; Wang et al., 2024) for fair comparison; further implementation details are provided in the Appendix. All experiments are conducted on four NVIDIA RTX 3090 GPUs.

### 5.1.2 EVALUATION PROTOCOLS

We adopt two complementary evaluation settings: **linear probing** and **image retrieval**. Linear probing is a widely used protocol in self-supervised learning (SSL). After pretraining, the backbone is frozen and a linear classifier is trained on top of the learned representations. The classification accuracy of this linear head serves as a direct measure of the discriminative quality of the features. Image retrieval evaluates how well the representations capture semantic similarity. For each query image, we perform nearest-neighbor search in the feature space and retrieve images from the gallery. Performance is measured by rank-$k$ accuracy, i.e., whether a correct match appears among the top-$k$ retrieved images. Unless otherwise specified, we report top-1 accuracy for linear probing and rank-1 / rank-5 accuracy for retrieval.

## 5.2 MAIN RESULTS

Table 1 reports a comprehensive comparison with prior methods on three fine-grained benchmarks. Our method consistently outperforms the baselines in both classification and retrieval. In particular, it surpasses LDF by **+3.45**, **+7.06**, and **+3.31** points in classification accuracy on CUB200, Cars, and Aircraft, respectively. Similar gains are observed in retrieval, where our approach achieves higher rank-1 and rank-5 accuracy across all datasets. These results highlight the robustness of our framework

Table 2: Classification accuracy (%) and retrieval performance (Rank-1 / Rank-5, %) for different ablation configurations of the proposed CLUE framework. The baseline (#1) uses standard contrastive learning with InfoNCE loss. Each row corresponds to incrementally adding key components: global loss (soft clustering alignment), local loss (part-aware contrastive learning), and text loss (VLM-guided semantic cues).

| ID | Losses | | | Classification | | | Retrieval | | | | | |
| | $L_{global}$ | $L_{local}$ | $L_{text}$ | CUB200 | Cars | Aircraft | CUB200 | | Cars | | Aircraft | |
| | | | | | | | Rank1 | Rank5 | Rank1 | Rank5 | Rank1 | Rank5 |
|---|---|---|---|---|---|---|---|---|---|---|---|---|
| #1 | | | | 62.29 | 60.20 | 51.13 | 37.74 | 57.07 | 31.74 | 56.05 | 31.74 | 52.07 |
| #2 | | | ✓ | 64.63 | 64.46 | 52.29 | 42.06 | 67.43 | 35.75 | 60.69 | 35.97 | 59.77 |
| #3 | ✓ | | | 65.82 | 64.76 | 56.70 | 42.53 | 71.31 | 34.92 | 60.46 | 34.64 | 58.84 |
| #4 | ✓ | ✓ | | 66.95 | 65.58 | 57.76 | 43.56 | 72.21 | 35.27 | 62.16 | 36.42 | 60.24 |
| #5 | ✓ | ✓ | ✓ | **69.62** | **72.66** | **58.59** | **48.53** | **74.71** | **43.45** | **69.49** | **40.66** | **63.56** |

Table 3: Fine-grained classification accuracy (%) and retrieval performance (Rank-1 / Rank-5, %) on CUB200 and FGVC-Aircraft for different number of clusters.

| Clusters | Classification | | | Retrieval | | | | | |
| | CUB200 | Cars | Aircraft | CUB200 | | Cars | | Aircraft | |
| | | | | Rank1 | Rank5 | Rank1 | Rank5 | Rank1 | Rank5 |
|---|---|---|---|---|---|---|---|---|---|
| 2 | 64.60 | 63.20 | 55.93 | 42.46 | 71.47 | 34.27 | 59.16 | 33.74 | 56.07 |
| 3 | 65.82 | 64.76 | 56.70 | 42.53 | 71.31 | 34.92 | 60.46 | 34.64 | 58.84 |
| 4 | 66.10 | 65.91 | 56.76 | 43.48 | 71.52 | 34.57 | 59.11 | 33.31 | 57.28 |
| 8 | 66.95 | 65.58 | 57.03 | 43.66 | 71.31 | 33.44 | 56.95 | 36.42 | 59.54 |

for both non-rigid objects (CUB200) and rigid categories (Cars, Aircraft), demonstrating its ability to capture subtle inter-class differences. Moreover, when instantiated with a ViT-Base backbone, our method further improves top-1 accuracy and consistently outperforms Transformer-based SSL approaches such as EsViT Li et al. (2021) and LoDisc Shi et al. (2025) in both classification and retrieval, confirming that the proposed multi-level regularization is architecture-agnostic and remains effective on modern vision Transformers.

## 5.3 ABLATION STUDY

**Effect of Key Modules.** We perform ablations on CUB200, Stanford Cars, and FGVC-Aircraft to isolate the contribution of each component (Table 2). The baseline (#1) relies on standard InfoNCE contrastive learning (Eq. equation 4). Replacing it with global soft clustering alignment (#3; soft-InfoNCE, Eq. equation 5) leads to clear gains in both classification and retrieval, highlighting the benefit of adapting to batch-level semantic structure. Adding the part-level loss (#4; Sec. 4.2) provides further improvements by leveraging local descriptors to capture fine-grained cues within coarse categories. Additionally, incorporating the VLM-guided objective (#5; Sec. 4.3) achieves the strongest results, showing that external semantic anchors offer complementary supervision. Taken together,

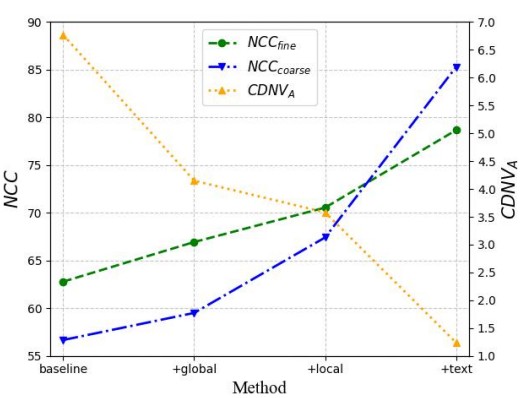

Figure 5: Clustering behavior on CIFAR-100.

these results demonstrate that global soft alignment, part-level discrimination, and VLM guidance contribute in a complementary and additive manner, producing the most effective fine-grained representations. Finally, when training with only the text loss (#2), we observe a noticeable degradation in linear probing performance, suggesting that VLM guidance alone is insufficient to learn strong fine-grained visual representations.

**Number of Clusters** In the part-assignment module, deep descriptors are grouped into $K$ clusters. As shown in Table 3, setting $K = 2$ typically separates foreground from background but fails to capture richer part-level semantics, leading to suboptimal performance. Increasing $K$ to 4 yields clear

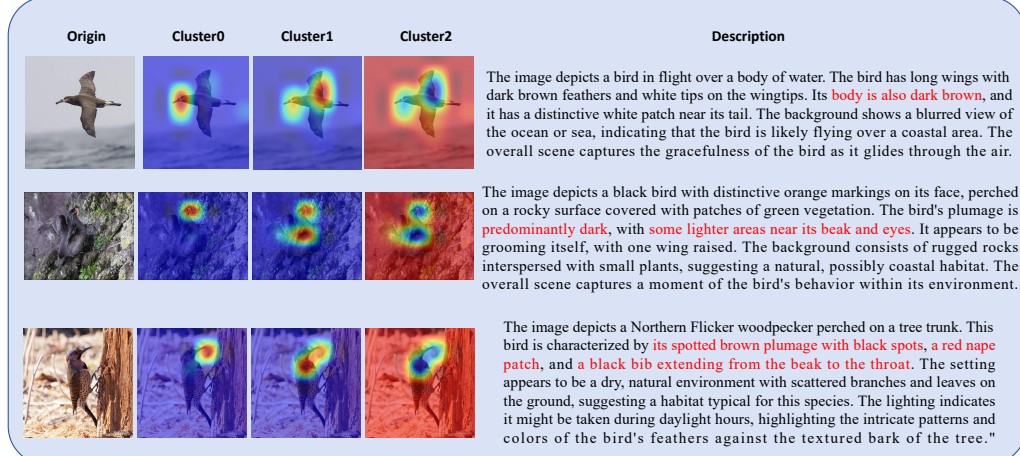

Figure 6: Visualization of the learned part clusters on fine-grained bird images. Each row shows the original image (left) and the response maps of three clusters (Cluster 0–2). Warmer colors indicate stronger activation of the corresponding cluster at that location. Cluster 0 consistently focuses on discriminative regions such as the head and beak, Cluster 1 attends to the torso/wing area, while Cluster 2 mainly captures background regions, illustrating that the part extractor discovers semantically meaningful parts.

improvements, indicating that moderate granularity helps model fine-grained cues. However, further enlarging $K$ does not consistently provide additional benefits and can even degrade results due to overfitting. Overall, the effect of cluster count tends to plateau beyond a moderate value. Balancing discriminative power and computational cost, we fix $K = 3$ for all experiments in this paper.

**Clustering Behavior of Module-Specific Clusters**    We evaluate clustering dynamics on CIFAR-100 using three granularity-aware metrics: (1) the average $\mathrm{CDNV}_A$ over the 20 coarse categories as an indicator of cluster compactness, (2) the average $\mathrm{NCC}_{\mathrm{fine}}$ over the 100 fine classes to measure fine-grained discrimination, and (3) the average $\mathrm{NCC}_{\mathrm{coarse}}$ over the 20 coarse categories to reflect super-class separability. An ideal outcome corresponds to a smaller $\overline{\mathrm{CDNV}_A}$ together with higher $\overline{\mathrm{NCC}_{\mathrm{fine}}}$, while maintaining $\overline{\mathrm{NCC}_{\mathrm{coarse}}}$. As shown in Figure 5, our module effectively suppresses cluster collapse and enhances fine-grained discrimination, without sacrificing coarse-level separability. This confirms its role in shaping balanced feature geometry across semantic granularities.

**The effect of the Part Extractor**    Our part extractor groups spatial features into clusters that correspond to semantically coherent regions in the image. To illustrate its behavior, we visualize several examples in Fig. 6. Even though no bounding-box annotations are used during training, the module automatically discovers meaningful parts: for instance, one cluster (Cluster 0) consistently focuses on the head and beak region, while another (Cluster 2) concentrates on the background. Similar patterns are observed across different categories, which aligns well with the common intuition in fine-grained recognition that stable, reusable parts are crucial for discrimination. In the visualizations, higher response ("hotter" colors) indicates stronger focus from the corresponding cluster on that region.

## 6 CONCLUSION

We studied the problem of fine-grained recognition in self-supervised learning, where standard objectives often suffer from over-dispersion or over-collapse of fine categories. To address this, we proposed a multi-level regularization framework that integrates soft-InfoNCE, part-aware learning, and VLM-guided alignment to shape feature geometry across granularities. Experiments on CUB200, Stanford Cars, and FGVC-Aircraft demonstrated consistent improvements in both classification and retrieval, with ablations confirming the complementary effect of each module. Our results highlight the importance of granularity-aware regularization for learning discriminative and transferable representations without labels.

ETHICS STATEMENT

This work does not involve human subjects, personally identifiable information, or sensitive data. All datasets used are publicly available, properly cited, and comply with their respective licenses. The proposed methodology is designed for fine-grained self-supervised learning research and does not introduce foreseeable risks of misuse or harmful societal impact. We have adhered to the ICLR Code of Ethics throughout the research and preparation of this paper.

REPRODUCIBILITY STATEMENT

To facilitate reproducibility of CLUE, we provide comprehensive dataset descriptions and full experimental details in in Appendix A, The supplementary materials include the complete source code, training scripts, and step-by-step instructions, covering data preprocessing, model configurations, and hyperparameters, enabling independent verification of all reported results.

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

# A  IMPLEMENTATION DETAILS

## A.1  DATASETS

We conducted experiments on widely used fine-grained recognition datasets to validate the effectiveness of our algorithm. Specifically, CUB-200-2011 consists of 11,788 images spanning 200 bird species, with 5,994 images allocated for training and 5,794 for testing. The Stanford Cars dataset comprises 16,185 images across 196 categories, divided into 8,144 training images and 8,041 testing images. FGVC-Aircraft includes 10,000 images covering 100 categories, with 6,667 images used for training and 3,333 reserved for testing.

For additional evaluation, we also consider four commonly used recognition benchmarks. The Oxford 102 Flowers dataset contains 8,189 images from 102 flower categories, following the standard split into training, validation, and test sets. The Oxford-IIIT Pet dataset includes 7,349 images of 37 breeds of cats and dogs, with roughly half of the images used for training and the remainder for testing. Food-101 consists of 101,000 images from 101 food categories, with 750 training images and 250 test images per class. Caltech-256 contains 30,607 images spanning 256 object categories, where we follow the conventional protocol and sample a fixed number of images per class for training while using the remaining images for testing.

## A.2  SETTINGS

For all our experiments, we use ResNet-50 (He et al., 2016) as the backbone, initialized with ImageNet-1K pre-trained weights, and employ a standard three-layer MLP as the projector. This initialization is kept consistent with prior fine-grained SSL works such as LCR(Shu et al., 2023) and PAPN(Tan et al., 2025): we use the official TensorFlow ResNet-50 checkpoint trained in a supervised manner on ImageNet-1K only.

Following common practice (He et al., 2020; Grill et al., 2020; Caron et al., 2020), we adopt a momentum encoder; in our main experiments the momentum coefficient is fixed to 0.999. To ensure a fair comparison, this choice is aligned with LCR(Shu et al., 2023) and LDF(Wang et al., 2024), which also use a momentum value of 0.999. For reference, other self-supervised methods such as ReSA(Weng et al., 2025) and DINO(Caron et al., 2021) decrease the effective update rate by gradually increasing the momentum from 0.996 to 1.0 over training. Since clustering in our framework is performed online, there is no need for a memory bank to store negatives as in MoCo. Instead, to ensure sufficient diversity within each batch, we set the batch size to 512, which is still much smaller than the queue size commonly used in MoCo-style methods. The number of prototypes for part-level clustering is set to $N = 3$.

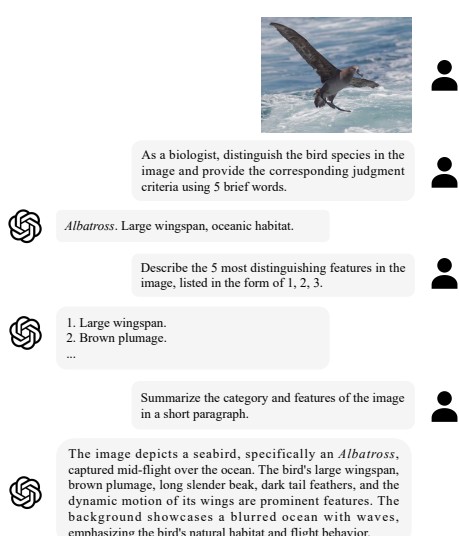

Figure 7: An example of generating a discriminative image description through a chain-of-thought dialogue with a vision-language model.

Based on the above settings, the only tuned hyperparameters are the optimizer-related ones, including the learning rate, weight decay, and the number of warm-up epochs, which are adjusted according to the specific encoder architecture and dataset; all other settings are kept fixed. We follow the linear scaling rule, setting the learning rate as $lr = lr_{base} \times batch\_size/256$. After the warm-up phase, the learning rate decays following a cosine schedule.

For the MAE baseline (He et al., 2022), we follow the official protocol, employing a ViT-Small backbone with a masking ratio of 75% and pre-training for 100 epochs to ensure a fair comparison. ViT-Small is chosen because its model size is comparable to that of ResNet-50. Unless otherwise specified, all methods share the same data preprocessing pipeline and optimization hyperparameters.

| Images | Descriptions |
|---|---|
| | The image depicts a bird swimming in water. The bird has a predominantly **brown plumage with lighter patches on its head and neck**. It possesses **a long, pointed beak** typical of seabirds adapted for catching prey. Its wings are **partially submerged**, indicating it is actively swimming or diving. The water around the bird shows ripples and splashes, suggesting movement. The background consists of a calm body of water with gentle waves, providing a natural aquatic habitat for the bird. |
| | The image depicts a vibrant red cardinal perched on a lichen-covered branch. The bird's striking plumage is characterized by its **bright red feathers, black mask around the beak**, and **a prominent crest atop its head**. The background is a soft green gradient, providing a natural contrast that highlights the cardinal's vivid coloration. The branch, adorned with patches of white lichen, adds texture to the scene, emphasizing the bird's delicate perch. This composition captures the essence of nature, showcasing the cardinal's beauty against a serene backdrop. |
| | The image depicts a bird perched on a tree branch against a clear blue sky. The bird has a predominantly **gray head with a black patch around its neck** and **a speckled pattern on its body**. Its **tail feathers are reddish-brown**, adding a striking contrast to its overall appearance. The background is slightly blurred, highlighting the bird as the focal point. The branches surrounding the bird have sparse leaves, suggesting it might be autumn or early spring. The lighting indicates that the photo was taken during daylight hours, possibly under direct sunlight. |

Figure 8: Visualization of discriminative descriptions identified by a vision-language model (VLM) on the CUB-200 dataset. The highlighted descriptions in red denote the most discriminative textual cues associated with the corresponding bird images.

For a fair comparison with LoDisc (ViT-B) (Shi et al., 2025), we adopt a ViT-Base backbone in our ViT-based experiments, matching the model size used in LoDisc. Likewise, both ViT-Small and ViT-Base are initialized from the official TensorFlow checkpoints pre-trained in a supervised manner on ImageNet-1K.

All experiments are conducted on 4 NVIDIA RTX 3090 GPUs.

## B  GENERATING DESCRIPTION WITH VLM

We explored using Vision-Language Models (VLMs) to generate discriminative descriptions. Inspired by the chain-of-thought approach, we adopted a multi-turn dialogue process for description generation. As illustrated in Figure 7, we first assign a specific role for each dataset, and then prompt the model from both global and local perspectives. Finally, a comprehensive textual description is generated for each image, ensuring appropriate length and detail.

After obtaining the generated feature descriptions, we utilize the text encoder of CLIP to project the textual information into the feature space. These projected text features serve as anchors, aiding the model's learning process.

Figure 8 presents example visualizations of discriminative descriptions generated by GLM-4V-flash on the CUB-200 dataset. Although GLM-4V-flash is not currently the most advanced vision-language model, it is nonetheless capable of effectively identifying and articulating distinctive image features.

## C  MORE EXPERIMENTS

To further validate the generality of CLUE, we additionally evaluate on three fine-grained benchmarks (Flowers, Pets, Food-101) and a more generic recognition benchmark (Caltech-256). For all these datasets, we strictly follow the training protocol described in the main text: the same ResNet-50 backbone initialized from the ImageNet-1K supervised checkpoint, identical data augmentations, batch size, number of epochs, and optimizer settings. The only change is the underlying dataset. For LCR, we use the official implementation and re-train under this unified setup to obtain the numbers reported in Tables 4 and 5.

Table 4: Fine-grained classification accuracy (%) and retrieval performance (Rank-1 / Rank-5, %) on Flowers, Pets, and Food-101.

| Method | Classification | | | Retrieval | | | | | |
| | Flower | Pet | Food | Flower | | Pet | | Food | |
| | | | | Rank1 | Rank5 | Rank1 | Rank5 | Rank1 | Rank5 |
|---|---|---|---|---|---|---|---|---|---|
| LCR (Shu et al., 2023) | 85.24 | 90.73 | 73.34 | 90.26 | 97.15 | 89.78 | 97.77 | 52.62 | 74.70 |
| ReSA (Weng et al., 2025) | 84.17 | 91.82 | 72.91 | 93.89 | 97.92 | 89.24 | 97.87 | 56.08 | 75.99 |
| CLUE (Ours) | 86.28 | 91.63 | 83.37 | 95.51 | 98.06 | 90.68 | 97.87 | 67.14 | 83.89 |

Table 5: Fine-grained classification accuracy (%) and retrieval performance (Rank-1 / Rank-5, %) on Caltech-256.

| Method | Classification | Retrieval | |
| | | Rank1 | Rank5 |
|---|---|---|---|
| LCR (Shu et al., 2023) | 74.44 | 75.82 | 87.85 |
| ReSA (Weng et al., 2025) | 84.07 | 78.36 | 88.04 |
| CLUE (Ours) | 86.39 | 80.22 | 90.90 |

The results show that CLUE consistently matches or outperforms strong baselines across all four datasets. On the fine-grained benchmarks in Table 4, CLUE achieves the best classification accuracy on Flowers (86.28%, +1.0 over LCR) and Food-101 (83.37%, more than +10 points over both LCR and ReSA), and also yields higher rank-1 retrieval scores on all three datasets, e.g., +14.5 points over LCR on Food-101. On Pets, CLUE attains competitive classification performance (91.63%) while still improving rank-1 retrieval over both LCR and ReSA. On the more generic Caltech-256 dataset (Table 5), CLUE significantly surpasses LCR (86.39% vs. 74.44% in classification, and 80.22% vs. 75.82% in rank-1 retrieval) and also improves over ReSA. These results indicate that the proposed multi-level regularization not only benefits fine-grained recognition but also transfers well to broader object recognition scenarios.

## D  GEOMETRIC IDENTITIES AND OPTIMIZATION VIEW OF CLUE

It is widely acknowledged that providing a rigorous theoretical explanation of how regularization terms induce clustered representations is challenging; (Ben-Shaul et al., 2023) have emphasized this difficulty in their discussions of representation learning. Inspired by prior geometric analyses of contrastive and prototype-based methods (Wang & Isola, 2020; Khosla et al., 2020; Snell et al., 2017), we take a modest step in this direction. In this appendix, we present a simple geometric view of how the proposed multi-level regularization shapes the feature space. We show that our objectives tend to (i) reduce intra-class variance and (ii) enlarge inter-class separation, thereby improving the CDNV metric used in the main text.

Throughout, we assume that feature vectors are $\ell_2$-normalized, i.e., $\|z_i\|_2 = 1$ for all $i$, which is standard in contrastive learning.

### D.1  INTRA-CLASS VARIANCE AND SIMILARITY

Let $C$ be a set of indices (e.g., a class or a cluster) with $|C| = n$ and centroid

$$\mu = \frac{1}{n} \sum_{i \in C} z_i.$$

Define the intra-class variance

$$\mathrm{Var}_{\mathrm{in}}(C) = \frac{1}{n} \sum_{i \in C} \|z_i - \mu\|_2^2.$$

**Lemma 1** (Variance–similarity identity). *For unit-norm vectors, the intra-class variance is*

$$\mathrm{Var}_{\mathrm{in}}(C) = 1 - \mathrm{Sim}_{\mathrm{in}}(C), \qquad \mathrm{Sim}_{\mathrm{in}}(C) = \frac{1}{n^2} \sum_{i,j \in C} \langle z_i, z_j \rangle. \tag{9}$$

*Proof.* Expanding the variance,

$$\text{Var}_{\text{in}}(C) = \frac{1}{n} \sum_{i \in C} (\|z_i\|_2^2 - 2\langle z_i, \mu \rangle + \|\mu\|_2^2).$$

Using $\|z_i\|_2^2 = 1$ and $\frac{1}{n} \sum_{i \in C} \langle z_i, \mu \rangle = \|\mu\|_2^2$ gives $\text{Var}_{\text{in}}(C) = 1 - \|\mu\|_2^2$. Finally,

$$\|\mu\|_2^2 = \left\langle \frac{1}{n} \sum_i z_i, \ \frac{1}{n} \sum_j z_j \right\rangle = \frac{1}{n^2} \sum_{i,j \in C} \langle z_i, z_j \rangle = \text{Sim}_{\text{in}}(C).$$

$\square$

Thus, on the unit sphere, maximizing intra-class similarity $\text{Sim}_{\text{in}}(C)$ is exactly equivalent to minimizing intra-class variance $\text{Var}_{\text{in}}(C)$.

## D.2 CLASS SEPARATION AND INTER-CLASS SIMILARITY

Consider two disjoint sets $C$ and $C'$ with $|C| = n$, $|C'| = n'$ and centroids

$$\mu = \frac{1}{n} \sum_{i \in C} z_i, \qquad \mu' = \frac{1}{n'} \sum_{j \in C'} z_j.$$

Define the inter-class similarity

$$\text{Sim}_{\text{out}}(C, C') \ = \ \frac{1}{nn'} \sum_{i \in C, \, j \in C'} \langle z_i, z_j \rangle.$$

**Lemma 2** (Centroid distance decomposition)**.** *The squared distance between centroids satisfies*

$$\|\mu - \mu'\|_2^2 \ = \ \text{Sim}_{\text{in}}(C) + \text{Sim}_{\text{in}}(C') - 2\,\text{Sim}_{\text{out}}(C, C'). \tag{10}$$

*Proof.* We have
$$\|\mu - \mu'\|_2^2 = \|\mu\|_2^2 + \|\mu'\|_2^2 - 2\langle \mu, \mu' \rangle.$$
Using $\|\mu\|_2^2 = \frac{1}{n^2} \sum_{i,j \in C} \langle z_i, z_j \rangle = \text{Sim}_{\text{in}}(C)$, similarly $\|\mu'\|_2^2 = \text{Sim}_{\text{in}}(C')$, and

$$\langle \mu, \mu' \rangle = \frac{1}{nn'} \sum_{i \in C, j \in C'} \langle z_i, z_j \rangle = \text{Sim}_{\text{out}}(C, C'),$$

we obtain equation 10. $\square$

This shows that increasing intra-class similarities $\text{Sim}_{\text{in}}(C)$, $\text{Sim}_{\text{in}}(C')$ and decreasing inter-class similarity $\text{Sim}_{\text{out}}(C, C')$ both enlarge the centroid distance.

## D.3 CONNECTION TO CDNV

For two sets $C$ and $C'$ with centroids $\mu, \mu'$, the pairwise CDNV metric (Eq. (1) in the main text) is

$$\text{CDNV}(C, C') \ = \ \frac{\text{Var}_{\text{in}}(C) + \text{Var}_{\text{in}}(C')}{2 \, \|\mu - \mu'\|_2^2}. \tag{11}$$

Assuming $\mu \neq \mu'$, Lemma 1 and Lemma 2 give

$$\text{CDNV}(C, C') \ = \ \frac{\big(1 - \text{Sim}_{\text{in}}(C)\big) + \big(1 - \text{Sim}_{\text{in}}(C')\big)}{2\Big(\text{Sim}_{\text{in}}(C) + \text{Sim}_{\text{in}}(C') - 2\,\text{Sim}_{\text{out}}(C, C')\Big)}. \tag{12}$$

Thus, strictly increasing $\text{Sim}_{\text{in}}(C)$, $\text{Sim}_{\text{in}}(C')$ and strictly decreasing $\text{Sim}_{\text{out}}(C, C')$ decreases the numerator and increases the denominator, and therefore strictly reduces $\text{CDNV}(C, C')$.

## D.4 OPTIMIZATION DYNAMICS OF SOFT-INFONCE

We now show how the Soft-InfoNCE loss used in CLUE acts on the pairwise similarities in a way that matches the CDNV analysis above.

Let $S \in \mathbb{R}^{m \times m}$ be a similarity matrix and define a row-wise softmax (for simplicity, without explicit temperature) by

$$D_{ij} = \frac{\exp(S_{ij})}{\sum_{k=1}^{m} \exp(S_{ik})}.$$

Let $A \in \mathbb{R}^{m \times m}$ be a row-stochastic target matrix ($\sum_j A_{ij} = 1$ for all $i$), and consider

$$L(S, A) = -\frac{1}{m} \sum_{i=1}^{m} \sum_{j=1}^{m} A_{ij} \log D_{ij}. \tag{13}$$

**Lemma 3** (Gradient of row-softmax cross-entropy). *For the loss $L(S, A)$ in equation 13,*

$$\frac{\partial L}{\partial S_{ij}} = \frac{1}{m}(D_{ij} - A_{ij}), \qquad i, j = 1, \ldots, m. \tag{14}$$

*Proof.* Writing $L = \frac{1}{m} \sum_i L_i$ with $L_i = -\sum_j A_{ij} \log D_{ij}$ and using

$$\log D_{ij} = S_{ij} - \log\left(\sum_k \exp(S_{ik})\right),$$

a standard calculation shows $\frac{\partial}{\partial S_{ij}} \log D_{ik} = \delta_{jk} - D_{ij}$, hence $\frac{\partial L_i}{\partial S_{ij}} = D_{ij} - A_{ij}$ and $\frac{\partial L}{\partial S_{ij}} = \frac{1}{m}(D_{ij} - A_{ij})$. $\square$

**Corollary 4** (Effect on pairwise similarities). *A gradient descent step $S_{ij} \leftarrow S_{ij} - \eta \, \partial L/\partial S_{ij}$ with $\eta > 0$ has:*

- *if $A_{ij} > D_{ij}$, then $\partial L/\partial S_{ij} < 0$ and $S_{ij}$ increases (the pair $(i, j)$ is pulled* closer*);*
- *if $A_{ij} < D_{ij}$, then $\partial L/\partial S_{ij} > 0$ and $S_{ij}$ decreases (the pair $(i, j)$ is pushed* apart*).*

In CLUE, $A$ is obtained from a Sinkhorn-based clustering and assigns larger mass to semantically related examples. Therefore the global and local regularization terms:

- increase similarities for pairs that should be close (increasing $\mathrm{Sim}_{\mathrm{in}}$ and reducing the CDNV numerator);
- decrease similarities for unrelated pairs (reducing $\mathrm{Sim}_{\mathrm{out}}$ and increasing the CDNV denominator).

By Lemma 1 and Lemma 2, this means that the global and local losses tend to reduce intra-class variance and enlarge inter-class distances, thus decreasing CDNV.

## D.5 PREVENTION OF TRIVIAL COLLAPSE VIA TEXT GUIDANCE

Finally, we show that a CLIP-style text alignment loss rules out the trivial solution where all image features collapse to the same constant vector.

Let $v_i$ be (normalized) image features and $t_i$ the corresponding text features. Consider the CLIP-style loss

$$L_{\text{text}} = -\frac{1}{m} \sum_{i=1}^{m} \log \frac{\exp(\langle v_i, t_i \rangle / \tau)}{\sum_{j=1}^{m} \exp(\langle v_i, t_j \rangle / \tau)}, \tag{15}$$

with temperature $\tau > 0$, and let

$$p_{ij} = \frac{\exp(\langle v_i, t_j \rangle / \tau)}{\sum_{k=1}^{m} \exp(\langle v_i, t_k \rangle / \tau)}$$

be the softmax probabilities.

**Lemma 5** (Constant mapping is not stationary). *Assume that the text features $\{t_i\}_{i=1}^m$ are not all equal. Then the mapping $v_1 = \cdots = v_m = c$ (for any constant c) is not a stationary point of $L_{\text{text}}$.*

*Proof.* The gradient w.r.t. $v_i$ is

$$\frac{\partial L_{\text{text}}}{\partial v_i} = \frac{1}{m\tau}\Big(\sum_{j=1}^m p_{ij}t_j - t_i\Big).$$

If $v_1 = \cdots = v_m = c$, then $p_{ij}$ does not depend on $i$, say $p_{ij} = p_j$, and

$$\frac{\partial L_{\text{text}}}{\partial v_i} = \frac{1}{m\tau}\Big(\sum_j p_j t_j - t_i\Big).$$

Stationarity would require $\sum_j p_j t_j = t_i$ for all $i$, i.e., all $t_i$ equal the same convex combination of $\{t_j\}$, which is impossible if the $\{t_i\}$ are not all identical. Hence at least one gradient is nonzero. $\square$

Therefore, the text-based regularization term explicitly rules out the trivial "all features are identical" solution and helps prevent extreme instance-level collapse, complementing the clustering-based global and local regularization discussed above.

## E  VISUALIZATION

For a clearer illustration of the model's effectiveness, we conduct Grad-CAM visualizations on the evaluation datasets, highlighting the regions most relevant to fine-grained discrimination  9.

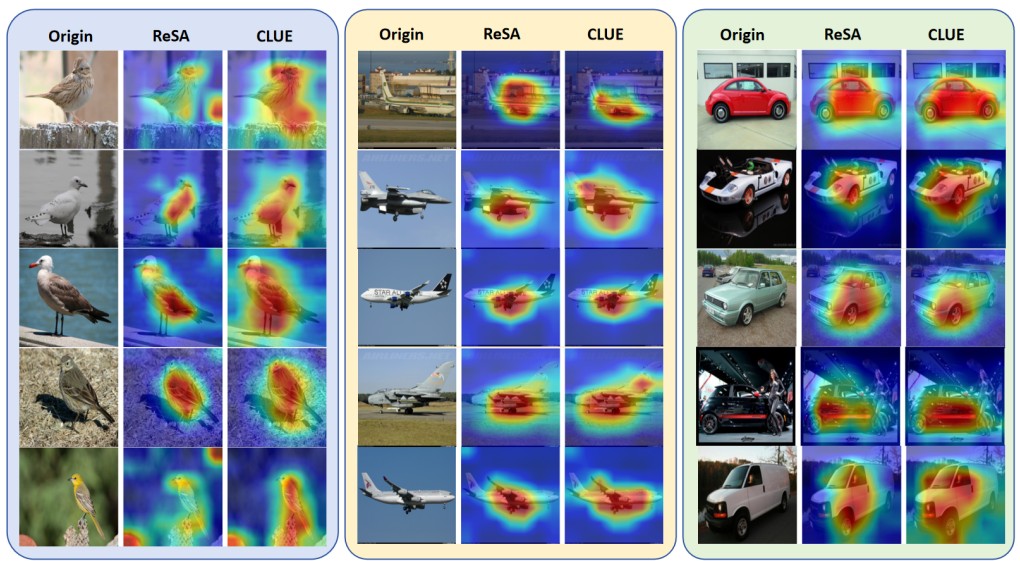

Figure 9: Grad-CAM visualizations on fine-grained benchmarks. Compared with baseline SSL, our CLUE model attends more accurately to discriminative regions (e.g., textures, shapes, or parts), which facilitates precise recognition and highlights its advantage in fine-grained categorization.

## F  THE USE OF LARGE LANGUAGE MODELS(LLMS)

Large Language Models (LLMs) were used only as language and formatting assistants during manuscript preparation. Specifically, LLMs were employed to (i) polish grammar and improve fluency, (ii) standardize terminology, tense, and voice, (iii) suggest alternative phrasings for clarity and concision, and (iv) provide suggestions for table layouts and LaTeX typesetting (e.g., caption style, column alignment, and cross-referencing).

Within our proposed framework, LLMs are included solely as supportive modules, functioning as tools to facilitate the architecture rather than as core research contributions.

LLMs did not participate in designing experiments, analyzing data, deriving theoretical results, or drawing conclusions. All technical ideas, methods, proofs, experimental protocols, and findings are authored, validated, and interpreted by the authors. All LLM-assisted edits were reviewed and approved by the authors to ensure accuracy and faithfulness to the intended meaning.

