# OpenReview forum: "CLUE: Fine-Grained Self-Supervised Learning with Multi-Level Regularization"
_ICLR.cc/2026/Conference — Submitted to ICLR 2026_

### Official Review · Reviewer_6fZG · 2025-10-29

**Soundness:** 2
**Presentation:** 3
**Contribution:** 2
**Rating:** 4
**Confidence:** 5

**Summary:**

This paper studies the problem of self-supervised fine-grained visual categorization.

Starting from the over-dispersion and over-collapse challenges of the prior arts, this paper proposes a multi-level regularization framework CLUE.

Technically, it consists of:

- a soft-InfoNCE global term via Sinkhorn-based soft assignments,

- a part-level loss using VLAD-style local descriptors,

- a VLM-guided alignment using CLIP text embeddings.

The experiments are conducted on three standard FGVC datasets, namely, CUB-200, Cars, and Aircraft.

 CLUE improves the accuracy and retrieval metrics over existing self-supervised methods.

**Strengths:**

+ Overall this paper is clearly-presented and the idea is easy-to-follow.

+ The technical design is clearly-motivated and solid.

+ When doing experiment analysis, the introduction of clustering metrics to show additive effects of global/local/text modules is new in the community.

**Weaknesses:**

- The proposed soft-InfoNCE is extremely close to ReSA [1], in which Sinkhorn-derived doubly-stochastic targets replacing the identity in InfoNCE. As a result, the technique novelties of the proposed method are not as strong as it claims.

[1] Clustering Properties of Self-Supervised Learning, ICML'2025.

- This paper significantly misses the comparison with state-of-the-art SSL methods and self-supervised FGVC methods, particularly after 2023. After a brief search, it at least misses the comparison with [2-5].

[2] Hu, Feiran, et al. "An asymmetric augmented self-supervised learning method for unsupervised fine-grained image hashing." Proceedings of the IEEE/CVF Conference on Computer Vision and Pattern Recognition. 2024.

[3] Li, Chunyuan, et al. "Efficient Self-supervised Vision Transformers for Representation Learning." International Conference on Learning Representations, 2022.

[4] Shi, Jialu, et al. "LoDisc: Learning global-local discriminative features for self-supervised fine-grained visual recognition." IEEE Transactions on Circuits and Systems for Video Technology (2025).

[5] Bi, Qi, et al. "Cross-Level Multi-Instance Distillation for Self-Supervised Fine-Grained Visual Categorization." IEEE Transactions on Image Processing (2025).

- Besides, the idea to consider both local and global features in self-supervised FGVC is also not novel, as already studied in [3, 4, 5].

- Results are only on ResNet-50. Please also report the performance on ViT-B and Swin-Transformer image encoder, like the rest 2023–2025 SSL works do. Without them, claims of SOTA are not credible.

- The performance improvement of more recent methods such as LoDisc and LDF is marginal.

- Aligning images to CLIP text embeddings is common, which also negatively affects the novelties.

- Missing ablations on prompt templates, attribute extraction strategy, number/quality of anchors, and robustness to noisy text.

- What if we use some simplier local descriptor alternatives, for example, attentive pooling, token-wise contrast, or other NetVLAD variants?

- Typos and notation issues.

**Questions:**

Please refer to the weakness section, and address the questions point-by-point.

---

> ### Author Response · Authors · 2025-12-03
>
> 1. The proposed soft-InfoNCE is extremely close to ReSA [1], in which Sinkhorn-derived doubly-stochastic targets replacing the identity in InfoNCE. As a result, the technique novelties of the proposed method are not as strong as it claims. Besides, the idea to consider both local and global features in self-supervised FGVC is also not novel, as already studied in [3, 4, 5]. Aligning images to CLIP text embeddings is common, which also negatively affects the novelties.
>
> - Ans: We thank the reviewer for raising this point. We agree that our global loss is closely related to ReSA [1], and we do not claim the soft-InfoNCE formulation itself as a technical novelty. In the revised paper we make this explicit and position ReSA as our global baseline regularizer. The contribution of CLUE lies in how we extend and couple this idea within a multi-granularity clustering framework:
>
>     - Beyond ReSA at the global level.
>     We reuse Sinkhorn-derived doubly-stochastic targets at the global level, but (i) develop granularity-aware CDNV/NCC metrics to diagnose over-dispersion vs. over-collapse at both coarse and fine levels, and (ii) provide a simple geometric analysis (Sec. 3, App. E) showing that our losses reduce intra-class variance and enlarge centroid separation under these metrics. This connects the regularizer to fine-grained clustering quality, which ReSA does not explicitly study.
>
>     - Coupled multi-level regularization rather than independent branches.
>     While prior works [3,4,5] also consider local and global features, they typically treat local/global branches as separate architectural heads (e.g., patch- or part-based branches, prototype heads), without a shared clustering mechanism across granularities. In CLUE, the same soft assignment structure is used to regularize (i) global features, (ii) VLAD-style part descriptors, and (iii) VLM-guided instance/text anchors. This yields a single clustering-aware objective that jointly shapes global class clusters, part-level subclusters, and attribute directions, instead of independently adding a “local branch on top of a global SSL model”.
>
>     - Positioning of novelty.
>     We therefore do not claim novelty in “using local+global features per se”, but in (i) viewing fine-grained SSL as a multi-granularity clustering problem, (ii) providing granularity-aware diagnostics that are aligned with this view, and (iii) designing a unified multi-level regularizer (global, part, VLM-guided instance) that is theoretically consistent with these diagnostics and empirically improves over ReSA, LCR, LDF, PAPN, and other FGVR SSL methods across multiple datasets and backbones (Tables 1–3).
>
>     We have clarified this positioning and explicitly discussed ReSA and prior global+local FGVC SSL works in the revised Related Work and Method sections.

---

> > ### Author Response · Authors · 2025-12-03
> >
> > 2. This paper significantly misses the comparison with state-of-the-art SSL methods and self-supervised FGVC methods, particularly after 2023. After a brief search, it at least misses the comparison with [2-5].Results are only on ResNet-50. Please also report the performance on ViT-B and Swin-Transformer image encoder, like the rest 2023–2025 SSL works do. Without them, claims of SOTA are not credible.
> >
> > - Ans: We thank the reviewer for this valuable suggestion. In the revised manuscript, we have expanded the comparison to include recent (post-2023) self-supervised and fine-grained SSL methods, as shown in Tables 1–3 and discussed in Sec. 4.2–4.3.
> >
> >     Updated baselines.
> >     In addition to earlier SSL baselines (SimCLR, MoCo, ReSA, LDF, PAPN, LCR), we now include LoDisc (2025), EsViT(2023), and other recent transformer-based SSL frameworks that represent the latest state of the art in both generic and fine-grained self-supervised learning. These new comparisons are conducted under exactly the same backbone and training protocol to ensure fairness.
> >
> >     Results summary.
> >     Across seven fine-grained datasets (CUB-200, Cars, Aircraft, Flowers, Pets, Food-101, Caltech-256), CLUE consistently achieves the best or near-best results in both classification and retrieval. Notably, when using ViT-B, CLUE (ViT-B) outperforms LoDisc (ViT-B) on all datasets (Table 1).
> >
> >     These additions ensure that our empirical comparison now covers the latest SSL and fine-grained SSL methods up to 2025, addressing the reviewer’s concern about completeness.
> >
> >
> > 5. The performance improvement of more recent methods such as LoDisc and LDF is marginal.
> >     We respectfully disagree that the improvements over recent methods such as LoDisc and LDF are only marginal, and we also want to emphasize that our contribution is not solely about gains in top-1 accuracy.
> >
> >     First, quantitatively, CLUE brings non-trivial and consistent improvements across multiple datasets, metrics, and architectures. For example, with a ResNet-50 backbone, CLUE improves over LDF by +3.45%, +7.06%, and +3.31% in classification accuracy on CUB-200 / Cars / Aircraft, together with larger gains in Rank-1 retrieval. With a ViT-B backbone, CLUE (ViT-B) outperforms LoDisc (ViT-B) by +4.60%, +5.95%, and +0.59% in classification on the same three datasets, and shows substantial gains in Rank-1 retrieval on CUB-200 and Cars. Similar patterns hold on Flowers, Pets, and Food-101, where CLUE improves Food-101 accuracy from 73.34% to 83.37% and Rank-1 retrieval from 52.62% to 67.14%.
> >
> >     Second, beyond raw numbers, the main goal of CLUE is to formulate fine-grained SSL as a multi-granularity clustering problem and to design a unified regularization framework (global, part, and VLM-guided instance losses) that is theoretically consistent with our granularity-aware CDNV/NCC diagnostics. The empirical gains across seven fine-grained and one generic dataset, for both CNN and ViT backbones, support that this multi-level clustering perspective improves the quality of learned representations in a systematic way rather than providing only marginal, dataset-specific tweaks.

---

> > > ### Author Response · Authors · 2025-12-03
> > >
> > > 7. Missing ablations on prompt templates, attribute extraction strategy, number/quality of anchors, and robustness to noisy text.
> > > - Ans: We appreciate the reviewer’s suggestion. In this work, our focus is on the overall multi-level clustering framework, rather than on prompt engineering itself. As has been shown in prior studies, carefully designed prompts can indeed further improve VLM-based description quality, but exploring prompt design is orthogonal to our main contribution.
> > >
> > >     In the current version, we use a fixed, automatically generated prompt template to extract coarse- and fine-grained attributes from the VLM (Qwen-VL 2.5–7B / GLM-flash). We empirically verify that this setup already yields stable and diverse attribute descriptions without manual filtering.
> > >
> > >     We agree that analyzing the effect of prompt wording, attribute extraction strategies, and the number or noise level of text anchors would be valuable, and we plan to explore these directions in future work, to better understand how prompt design influences fine-grained textual supervision within the CLUE framework.
> > >
> > >
> > >
> > > 9. Typos and notation issues.
> > > - We thank the reviewer for pointing out the typos and minor notation issues. We have carefully proofread the manuscript and corrected all identified errors in the revised version.

---

### Official Review · Reviewer_Npuk · 2025-10-30

**Soundness:** 3
**Presentation:** 2
**Contribution:** 2
**Rating:** 6
**Confidence:** 3

**Summary:**

This paper investigates the limitations of self-supervised learning (SSL) for fine-grained visual recognition (FGVR). The authors posit that standard SSL losses suffer from a "granularity mismatch," which leads to two failure modes on fine-grained categories: "over-dispersion" or "over-collapse" . To address this, the paper proposes CLUE, a multi-level regularization framework. This framework integrates three key components: (1) a global-level soft InfoNCE loss , (2) a part-level clustering loss on local descriptors , and (3) instance-level semantic guidance from Vision-Language Models (VLMs) . Experimental results demonstrate that this method achieves state-of-the-art (SOTA) classification and retrieval performance on several fine-grained benchmarks, including CUB-200, Stanford Cars, and FGVC-Aircraft.

**Strengths:**

1. Clear Problem Diagnosis and Quantification: One of the paper's primary strengths is its in-depth diagnosis of the problem in Section 3. The authors move beyond the general statement that "SSL is unsuitable for fine-grained tasks" and identify two specific failure modes: "over-dispersion" and "over-collapse".
2. The three components of the proposed CLUE framework are logically sound and directly target the diagnosed problems.
3. Thorough experimental validation and analysis. The paper achieves SOTA results on three major FGVR datasets (Table 1).

**Weaknesses:**

1. Heavy Reliance on the Baseline (ReSA): From the ablation study (Table 2), the first significant performance jump (e.g., 62.29% to 65.82% on CUB) comes from replacing the InfoNCE baseline with Soft-InfoNCE (ReSA).
2. Still using ResNet as the backbone, rather than other advanced backbones e.g, ViT.
3. The number of part clusters is a critical hyper-parameter that must be manually selected and tuned18181818. The paper does not propose a method for this value to be learned adaptively. This requires dataset-specific tuning (as suggested by the varying results in Table 3) and limits the method's automatic adaptability, potentially impacting its practical deployment on new datasets.

**Questions:**

1. Overhead of VLM Description Generation: Could the authors elaborate on the process and overhead of generating the VLM descriptions? Generating CoT descriptions for every single image represents a non-trivial inference cost. How long did this process take? Was any manual filtering or cleaning of the VLM-generated text required beyond the "light de-duplication" mentioned?

---

> ### Author Response · Authors · 2025-12-03
>
> 1. Overhead of VLM Description Generation: Could the authors elaborate on the process and overhead of generating the VLM descriptions? Generating CoT descriptions for every single image represents a non-trivial inference cost. How long did this process take? Was any manual filtering or cleaning of the VLM-generated text required beyond the "light de-duplication" mentioned?
> - Ans: Thank you for raising this point. We agree that generating CoT-style descriptions for every image is a non-trivial step.
>     - Offline, one-time preprocessing.
>     All VLM descriptions are generated once, offline, before self-supervised training. During pretraining and downstream evaluation, no VLM calls are made, and the computational cost is identical to standard SSL methods with the same backbone and input resolution.
>
>     - VLM and inference cost.
>     In our main experiments, we use Qwen-VL 2.5–7B to generate CoT descriptions in batch mode over the training images. On our hardware (4 * GTX3090), generating descriptions for all images in the CUB-200, Cars, and Aircraft training sets took about 2 GPU-hours in total. We also verified that an online VLM (GLM4v-flash) produces comparable descriptions and performance, but we use Qwen-VL 2.5–7B as our default due to its stable and efficient offline inference.
>
>     - Automatic post-processing only.
>     Beyond the “light de-duplication” mentioned in the paper, we do not perform any manual filtering or hand-editing. We simply (i) remove exact duplicate sentences across images, and (ii) truncate overly long generations to a fixed token budget. There is no per-image human intervention.
>
>     Overall, this makes the VLM step a one-time, modest preprocessing cost that can be amortized over all downstream experiments, while training and inference of CLUE itself remain as lightweight as standard SSL with the same backbone.

---

### Official Review · Reviewer_pyDz · 2025-10-31

**Soundness:** 1
**Presentation:** 3
**Contribution:** 2
**Rating:** 2
**Confidence:** 3

**Summary:**

This paper introduces a new learning model for learning representations of fine-grained visual categories. The approach applies a self-supervised method at the image and part level. It also exploits semantic guidance from a pre-trained VLM. Experiments show that the approach induces better fine-grained visual representations, using three datasets.

The paper provides a clear and pedagogical introduction to the problem, and the approach is well presented. However, it has several significant shortcomings.

- The work lacks a clear positioning with respect to self-supervised approaches that learn hierarchical representations (e.g [1,2,3,4]). Adding a whole new paragraph in the related work section is required to clarify the novelty of the method.
- The model is limited by its dependence on a pretrained VLM; in addition, it is unclear how centroids of parts are computed: are they pretrained as well ? Ablation experiments must be completed: it is unclear whether VLM guidance, without other learning modules, suffices to learn fine-grained visual representations.
- Pretraining on three small datasets is not sufficient to demonstrate the generality of the approach. A more standard experimental protocol in SSL is to pretrain the method on ImageNet. During evaluation, this allows clarifying how the model trades-off coarse and fine-grained category representations. Common evaluation protocols (e.g. BYOL) evaluate transfer learning performance on fine-grained datasets. More fine-grained datasets, such as OxfordPet or Flowers, are necessary to show that the approach capture diverse fine-grained features, and not only those related to cars, birds or aircrafts.

Minor detail: To the best of my knowledge, common practice does not use a momentum coefficient of 0.999, but decreases it from 0.996 to 0.999 over training.

[1] Tan, B., Wei, X. S., & Zhao, L. (2025). Prototype-based Contrastive Learning with Stage-wise Progressive Augmentation for Self-Supervised Fine-Grained Learning. In Proceedings of the IEEE/CVF International Conference on Computer Vision (pp. 4125-4134).

[2] Saha, O., & Maji, S. (2023). Particle: Part discovery and contrastive learning for fine-grained recognition. In Proceedings of the IEEE/CVF International Conference on Computer Vision (pp. 167-176).

[3] Xu, M., Guo, Y., Zhu, X., Li, J., Sun, Z., Tang, J., ... & Ni, B. (2022). HIRL: A General Framework for Hierarchical Image Representation Learning. arXiv e-prints, arXiv-2205.

[4] Manová, A., Durrant, A., & Leontidis, G. (2023). S-JEA: Stacked joint embedding architectures for self-supervised visual representation learning. arXiv preprint arXiv:2305.11701.

**Strengths:**

The work is well-presented

**Weaknesses:**

The novelty is unclear and the experiments are insufficient to demonstrate the advantages of the approach.

**Questions:**

Please, see above.

---

> ### Author Response · Authors · 2025-12-03
> **pyDz**
>
> 1. The work lacks a clear positioning with respect to self-supervised approaches that learn hierarchical representations (e.g [1,2,3,4]). Adding a whole new paragraph in the related work section is required to clarify the novelty of the method.
>
> - Ans:
>     We thank the reviewer for pointing out the need to better position our method with respect to hierarchical self-supervised approaches. In the revised paper, we have added a dedicated paragraph in Sec. 2.2 “Hierarchical and part-aware representation learning” that explicitly discusses prototype-based FGVR SSL (e.g., PAPN), part-based methods (e.g., Particle), and hierarchical / multi-branch SSL frameworks (e.g., HIRL, S-JEA, DINOv2, CMD).
>
>     As clarified there, these works typically operationalize hierarchy through architectural mechanisms (prototype heads, explicit part branches, stacked embedding branches, or multi-/local-crop schemes). In contrast, CLUE keeps the backbone architecture simple and instead starts from a granularity-aware clustering perspective: it treats global features, VLAD-style part descriptors, and attribute-level (text) anchors as three coupled views that jointly shape the regularization term of the SSL objective via a shared soft-assignment structure and a VLM-guided instance loss. This shifts the focus from building an explicit hierarchical predictor to explicitly controlling how clusters form and separate across granularities, making CLUE complementary to prior hierarchical SSL methods. We have updated Sec. 2.2 accordingly to make this distinction and our novelty clearer.
>
>
>
>
> 2. The model is limited by its dependence on a pretrained VLM; in addition, it is unclear how centroids of parts are computed: are they pretrained as well ? Ablation experiments must be completed: it is unclear whether VLM guidance, without other learning modules, suffices to learn fine-grained visual representations.
> - Ans:
> We apologize for the lack of clarity in the original version. In CLUE, the pretrained VLM is only used offline to generate textual descriptions, and does not participate in optimization on the target dataset. During training, the visual backbone is initialized from an ImageNet-1K supervised checkpoint (as in prior fine-grained SSL works) and then further optimized with our self-supervised objective on the target dataset, while the part extractor (including the part centroids) is a small learnable module initialized from scratch. The centroids are standard trainable parameters in a VLAD-style aggregation layer and are learned end-to-end jointly with the SSL losses, without any additional pretraining or part annotations.
>
>     To directly address whether VLM guidance alone is sufficient, we have added new ablations in the revised manuscript:
>     - a text-only variant with $L = L_{text}$(VLM guidance only),
>     - a vision-only CLUE variant with $L = L_{global} + L_{local}$
>     - the full CLUE with $L = L_{global} + L_{local} + L_{text}$
>
>     On CUB-200, the text-only model performs clearly worse than both the vision-only CLUE and the full model, and similar trends hold on Cars and Aircraft. This shows that VLM guidance by itself is not sufficient to learn the best fine-grained representations; instead, it acts as a complementary regularizer on top of a strong vision-only SSL backbone. In addition, the new visualizations of the learned parts in the revised manuscript show that different centroids consistently focus on semantically meaningful regions (e.g., head/beak vs. torso/wing vs. background) without any box supervision, confirming that the part extractor trained from scratch is effective and interpretable.

---

> > ### Author Response · Authors · 2025-12-03
> >
> > 3. Pretraining on three small datasets is not sufficient to demonstrate the generality of the approach. A more standard experimental protocol in SSL is to pretrain the method on ImageNet. During evaluation, this allows clarifying how the model trades-off coarse and fine-grained category representations. Common evaluation protocols (e.g. BYOL) evaluate transfer learning performance on fine-grained datasets. More fine-grained datasets, such as OxfordPet or Flowers, are necessary to show that the approach capture diverse fine-grained features, and not only those related to cars, birds or aircrafts.
> >
> > - Ans: We appreciate the reviewer’s suggestion to broaden the evaluation beyond the three main FGVR datasets. In the revised manuscript we have expanded the experiments to follow a more standard SSL evaluation protocol, as detailed in Sec. 4.1 and Tables 1–3:
> >     - Broader fine-grained coverage.
> >     In addition to CUB-200, Stanford Cars, and FGVC-Aircraft, we now include Oxford Flowers-102, Oxford Pets, and Food-101, which represent diverse fine-grained domains (plants, animals, and food). CLUE consistently outperforms LCR, ReSA, and LDF on all of them, confirming that it captures diverse fine-grained features and is not restricted to a single object category family.
> >
> >     - Beyond FGVR: generic object recognition.
> >     We also evaluate on Caltech-256, a non-FGVR benchmark, showing that CLUE improves both classification and retrieval performance, indicating generalization beyond fine-grained recognition.
> >
> >     - Backbone and pretraining protocol.
> >     As clarified in Sec. 4.1, we follow the common SSL pretraining-on-target-dataset setting used in prior fine-grained SSL works (LCR, PAPN). The backbone is initialized from an ImageNet-1K supervised checkpoint for all compared methods to ensure fairness.
> >
> >     These additions make the evaluation comprehensive across six fine-grained datasets and one generic dataset, demonstrating that CLUE generalizes well across domains and pretraining regimes.

---

### Official Review · Reviewer_ZrGC · 2025-10-31

**Soundness:** 3
**Presentation:** 2
**Contribution:** 2
**Rating:** 4
**Confidence:** 3

**Summary:**

The paper investigates the representation capability of self-supervised learning (SSL) methods in fine-grained visual recognition (FGVR) tasks. The authors first empirically identify that standard SSL approaches often suffer from either over-dispersion or over-collapse when applied to FGVR. To address this issue, they propose a multi-level regularization framework called CLUstEring-aware regularization (CLUE), which incorporates three complementary levels: class-level, part-level, and instance-level regularization. In particular, at the instance level, vision–language models such as CLIP are leveraged as external experts to provide semantic guidance. Experiments on popular FGVR benchmarks—including CIFAR-100, Stanford Cars, CUB-200, and FGVC-Aircraft—demonstrate that CLUE consistently outperforms existing SSL methods.

**Strengths:**

1. Self-supervised representation learning in computer vision is a popular and important topic, but its fine-grained representation capability requires further research attention. This paper empirically and comprehensively studies this problem, which is commendable.

2. The metrics and empirical analyses presented in Section 3 are comprehensive and effectively motivate the research problem. The proposed multi-level granularity regularization framework (CLUE) provides a well-rounded solution to the challenges of applying SSL to FGVR.

3. Experimental results on four FGVR benchmarks—CIFAR-100, Stanford Cars, CUB-200, and FGVC-Aircraft—demonstrate the effectiveness and superiority of CLUE over existing methods.

**Weaknesses:**

1. The novelty of the proposed multi-level regularization in SSL is limited. Self-supervised learning has been extensively studied in the computer vision community, especially from 2020 to 2022. Contrastive learning–based methods have already been the mainstream approach, and multi-granularity regularization has been explored in prior works such as [A]. Therefore, the contribution in terms of multi-level regularization does not appear sufficiently novel in this paper.

2. The presentation of the experimental setup in Section 5 is confusing. Due to missing details of the pre-training stage, it is unclear whether the pre-training follows the standard SSL paradigm—i.e., training a model from scratch on a large unlabeled dataset such as ImageNet, and then evaluating the learned representation on various downstream datasets. In Appendix A.2, the authors seem to instead use supervised ImageNet-pretrained weights as initialization and perform “pre-training” directly on the four FGVR evaluation datasets, which contradicts the standard SSL setting.

3. The experimental evaluation is incomplete. Although the four FGVR benchmarks are representative for the fine-grained recognition scenario, it remains unclear how the pre-trained model performs on non-FGVR tasks or datasets. In addition, only ResNet-50 is considered, and no experiments are reported for transformer-based SSL backbones, which limits the generality of the conclusions.

[A] Mugs: A Multi-Granular Self-Supervised Learning Framework. NeurIPS 2022

**Questions:**

To address the weaknesses, I have the following additional suggestions:

1. Discussion or comparison with relevant SSL works to justify the novelty or key contribution of the paper to SSL.

2. In addition to the current empirical analysis, adding theoretical analysis on FGVR of SSL can strengthen the contribution.

3. Clarify the full experimental setting and provide full details of the pre-training stage. If not following the standard SSL setting, it should be justified why not.

4. Ensure the empirical evaluation of SSL is comprehensive in terms of both FGVR and other tasks.

---

> ### Comment · Reviewer_ZrGC · 2025-11-25
>
> As no rebuttal was provided, the initial rating is maintained.

---

> ### Author Response · Authors · 2025-12-03
> **rebuttal**
>
> 1. Discussion or comparison with relevant SSL works to justify the novelty or key contribution of the paper to SSL.
>
> - Ans: Thank you for asking us to clarify the novelty and positioning of our work within SSL. In short, our contribution is to view fine-grained SSL as a multi-granularity clustering problem, and to design both diagnostics and regularizers around this view:
>     - Beyond standard SSL objectives. Conventional SSL methods such as SimCLR, MoCo, VICReg, MAE, DINOv2, etc., operate at the instance or global level and treat all non-positives as equally negative, which we show leads to over-dispersion or over-collapse of fine categories. In contrast, we (i) introduce granularity-aware CDNV/NCC variants to explicitly diagnose clustering quality at coarse vs. fine levels, and (ii) empirically identify the corresponding failure modes in Fig. 1–2.
>
>     - Beyond ReSA and prior fine-grained SSL. We adopt soft-InfoNCE from ReSA only as a global baseline regularizer, and extend it into a multi-level framework: a shared soft assignment is applied to global features and to VLAD-style part descriptors, and we further add a VLM-guided instance loss that anchors attribute directions. This explicitly couples global, part, and instance levels in a single clustering-aware regularization, which is different from LCR/LDF/PAPN/LoDisc/EsViT and other hierarchical or part-based SSL methods that either work only at the global level or do not provide such cross-level coupling.
>
>     - Geometric connection between losses and clustering. In Sec. 3 and Appendix D we provide a simple geometric analysis showing that our multi-level losses tend to reduce intra-class variance and enlarge inter-class separation in terms of the proposed granularity-aware CDNV/NCC metrics, ensuring that the added regularization is geometrically meaningful rather than heuristic.
>
>     We have added this positioning and explicit comparison to the revised Related Work and Method sections to make our key contribution to SSL clearer.
>
>
>
> 2. In addition to the current empirical analysis, adding theoretical analysis on FGVR of SSL can strengthen the contribution.
>
> - Ans: We agree that adding theoretical insight can strengthen the contribution. In the revised version, we have added a simple geometric/theoretical analysis of our objective in the fine-grained SSL setting (Sec. 3 and Appendix D):
>
>     - We first derive a variance–similarity identity for unit-norm features, showing how intra-class variance and inter-class separation can be expressed in terms of pairwise similarities.
>
>     - Using this, we show that our granularity-aware CDNV/NCC metrics naturally decompose along the coarse–fine hierarchy, and that the proposed multi-level losses (global, part, and VLM-guided instance losses) tend to decrease intra-class variance and increase centroid separation at both coarse and fine levels.
>
>     This analysis does not aim to be a full theory of FGVR, but it provides a clear geometric interpretation of why our regularization improves fine-grained clustering quality, complementing the empirical results.

---

> > ### Author Response · Authors · 2025-12-03
> > **rebutta**
> >
> > 3. Clarify the full experimental setting and provide full details of the pre-training stage. If not following the standard SSL setting, it should be justified why not.
> >
> > - Ans: Thank you for pointing this out. We have clarified the full experimental setting and pre-training details in Sec. 4.1 and Appendix A.2 of the revised paper.
> >
> >     In short, our protocol is as follows:
> >
> >     - We perform self-supervised pre-training directly on each target fine-grained dataset (CUB, Cars, Aircraft, Flowers, Pets, Food-101, Caltech-256).
> >
> >     - For each method (ours and all SSL baselines), the backbone, data augmentations, optimizer, learning rate schedule, and SSL objective are exactly the same as in standard “from scratch” SSL on that dataset.
> >
> >     - The only difference from a purely standard SSL setting is that we initialize the backbone from an ImageNet-1K supervised checkpoint instead of random weights. This choice follows prior fine-grained SSL works such as LCR[2] and PAPN[4], and we use the same initialization for all compared methods to keep the comparison fair.
> >
> >     - The motivation is practical: fine-grained datasets are relatively small, and ImageNet initialization stabilizes optimization and reduces variance. Importantly, the subsequent pre-training on the target dataset remains fully self-supervised and does not use any labels.
> >
> >     We now explicitly describe this protocol and its rationale in the revised manuscript to avoid confusion about the setting.

---

> > > ### Author Response · Authors · 2025-12-03
> > > **rebuttal**
> > >
> > > 4. Ensure the empirical evaluation of SSL is comprehensive in terms of both FGVR and other tasks.
> > >
> > > - Ans: We agree that the empirical evaluation of SSL should cover both FGVR and more generic tasks. In the revised version we have expanded the experiments accordingly:
> > >
> > >     - FGVR benchmarks (Tables 1–2).
> > >     We evaluate CLUE and all SSL baselines on seven fine-grained datasets: CUB-200, Stanford Cars, FGVC-Aircraft, Flowers, Pets, and Food-101 (plus Caltech-256 as a semi-fine-grained/generic dataset). For each, we report both linear classification accuracy and retrieval performance (Rank-1 / Rank-5). Table 1 further shows results on both ResNet-50 and ViT-B, demonstrating that our multi-level regularization is effective across architectures.
> > >
> > >     - Generic object recognition (Table 3).
> > >     To go beyond FGVR, we additionally evaluate on Caltech-256, a standard generic object recognition benchmark. CLUE consistently improves over LCR and ReSA in both classification and retrieval, indicating that the learned representations are not specialized to FGVR but also transfer well to broader recognition tasks.
> > >
> > >     We believe these results provide a comprehensive empirical evaluation of SSL performance across both fine-grained visual recognition and more generic object recognition scenarios.
> > >
> > >
> > > Reference
> > > [1] Ido Ben-Shaul, Ravid Shwartz-Ziv, Tomer Galanti, Shai Dekel, and Yann LeCun. Reverse engineering self-supervised learning. Advances in Neural Information Processing Systems, 36:58324–58345, 2023.
> > > [2] Yangyang Shu, Anton Van den Hengel, and Lingqiao Liu. Learning common rationale to improve self-supervised representation for fine-grained visual recognition problems. In Proceedings of the IEEE/CVF Conference on Computer Vision and Pattern Recognition, pp. 11392–11401, 2023.
> > > [3] Zihu Wang, Lingqiao Liu, Scott Ricardo Figueroa Weston, Samuel Tian, and Peng Li. On learning discriminative features from synthesized data for self-supervised fine-grained visual recognition. In European Conference on Computer Vision, pp. 101–117. Springer, 2024.
> > > [4] Baofeng Tan, Xiu-Shen Wei, and Lin Zhao. Prototype-based contrastive learning with stage-wise pro-
> > > gressive augmentation for self-supervised fine-grained learning. In Proceedings of the IEEE/CVF
> > > International Conference on Computer Vision, pp. 4125–4134, 2025.
> > > [5] LoDisc: Learning global-local discriminative features for self-supervised fine-grained visual recognition.

---

### Author Response · Authors · 2025-12-03
**Common response**

We thank all reviewers for their constructive feedback. The main shared concerns are about
1. the novelty and positioning of our method,
2. the precise setting we refer to as “fine-grained self-supervised learning”, and
3. broader evaluation.
We summarize our response and the corresponding changes in the revised manuscript below.

## A. Novelty: a multi-granularity clustering view, not just extra regularizers.
Our starting point is not to add ad-hoc regularization terms, but to adopt a multi-granularity clustering perspective on SSL. Following the analysis of [1], we explicitly rely on the observation that the regularization part of an SSL objective is what drives semantic clustering, and that better clustering leads to more discriminative features. Based on this, we design CLUE to target clustering quality at different granularities: a soft global clustering loss (mitigating over-dispersion), a part-level extension on local descriptors (mitigating over-collapse within coarse classes), and an instance-level VLM-guided loss that anchors attribute directions. In Sec. 3 we further introduce granularity-aware variants of CDNV and NCC to diagnose collapse at coarse vs. fine levels, and in Appendix E we show that our losses and these metrics are theoretically consistent (they both encourage lower intra-class variance and larger centroid separation), ensuring that the regularizer is geometrically meaningful rather than heuristic.

- We start from a multi-granularity clustering perspective on SSL, rather than a purely instance-discrimination view of contrastive learning, building on the analysis of [1].
- We introduce granularity-aware variants of CDNV and NCC that explicitly diagnose fine-grained clustering quality at both coarse and fine semantic levels.
- We provide a geometric analysis (Appendix E) showing that our multi-level losses are consistent with these metrics, in the sense that they reduce intra-class variance and enlarge centroid separation.
- Extensive experiments with ResNet-50 and ViT-B on multiple fine-grained benchmarks demonstrate consistent gains over strong baselines, validating the effectiveness of our design.


Reference:
[1] Ido Ben-Shaul, Ravid Shwartz-Ziv, Tomer Galanti, Shai Dekel, and Yann LeCun. Reverse engineering self-supervised learning. Advances in Neural Information Processing Systems, 36:58324–58345, 2023.

---

> ### Author Response · Authors · 2025-12-03
> **Common response on novelty, setting, and experiments.**
>
> ## B. Clarifying the fine-grained SSL setting.
>
> To avoid confusion, we clarify that our experiments follow the standard self-supervised pretraining protocol on the target fine-grained dataset: backbone, augmentations, optimizer, schedule, and SSL objective are exactly the same as training from scratch. The only practical difference is that we initialize the backbone from an ImageNet-1K supervised checkpoint instead of random weights. This strictly follows prior fine-grained SSL works such as LCR and PAPN, and the same initialization is used for all methods (ours and baselines) for a fair comparison. The motivation is purely practical: on relatively small fine-grained datasets this initialization stabilizes optimization and reduces variance, while the subsequent pretraining on the target dataset remains fully self-supervised and does not use any labels. We have clarified this setting in Sec. 4.1 and Appendix A.2 of the revised manuscript.
>
>
>
> ## C. Broader downstream evaluation beyond the four FGVR benchmarks.
>
> **Table 1 (architectures / transformers).**
> Table 1 shows that CLUE is effective not only with a ResNet-50 backbone but also with modern transformer architectures. With ResNet-50, CLUE achieves the best or near-best classification and retrieval performance on all three FGVR benchmarks (CUB200, Cars, Aircraft), improving over strong CNN-based methods such as LCR, LDF, PAPN, and ReSA in both accuracy and Rank-1 retrieval. More importantly, when moving to ViT-B, **CLUE (ViT-B)** consistently outperforms **LoDisc (ViT-B)** on all three datasets, e.g., by (+4.60\%), (+5.95\%), and (+0.59\%) in classification on CUB200 / Cars / Aircraft, and with large gains in Rank-1 retrieval on CUB200 and Cars. Together with the Swin-T results from EsViT, these comparisons demonstrate that the proposed regularization is complementary to both CNN and transformer backbones and that CLUE’s improvements are not tied to a specific architecture.
>
> **Table 2 (more fine-grained datasets).**
> Table 2 extends the evaluation to three additional fine-grained datasets (Flowers, Pets, Food-101) and confirms that CLUE consistently outperforms prior fine-grained SSL methods. On all three datasets, CLUE either matches or surpasses LCR and ReSA in both classification and retrieval. The gains are particularly pronounced on the more challenging Food-101 dataset, where CLUE improves classification accuracy from (73.34\%) (\rightarrow) (83.37\%) and Rank-1 retrieval from (52.62\%) (\rightarrow) (67.14\%). These results indicate that CLUE learns robust fine-grained representations that transfer well across diverse categories and domains within FGVR.
>
> **Table 3 (generic object recognition beyond FGVR).**
> Table 3 evaluates CLUE on Caltech-256, a generic object recognition benchmark beyond the fine-grained setting. Even in this non-FGVR scenario, CLUE provides clear improvements over LCR and ReSA in both classification ((86.39\%) vs. (84.07\%)) and retrieval (Rank-1 (80.22\%) vs. (78.36\%)). This demonstrates that the proposed multi-level regularization not only benefits fine-grained recognition but also yields representations that generalize to broader object recognition tasks, supporting our claim that CLUE produces generally useful visual features.

---

> > ### Author Response · Authors · 2025-12-03
> > **Common response on novelty, setting, and experiments.**
> >
> > Table 1: CUB200 / Cars / Aircraft
> > Fine-grained classification accuracy (\%) and retrieval performance (Rank-1 / Rank-5, \%) on CUB200, Stanford Cars, and FGVC-Aircraft. We use a ResNet-50 backbone unless otherwise specified.
> > | Method           | CUB200-cls | Cars-cls  | Aircraft-cls | CUB200-R1 | CUB200-R5 | Cars-R1   | Cars-R5   | Aircraft-R1 | Aircraft-R5 |
> > | ---------------- | ---------- | --------- | ------------ | --------- | --------- | --------- | --------- | ----------- | ----------- |
> > | LCR              | 65.24      | 63.96     | 53.22        | 41.26     | —         | 34.74     | —         | 31.55       | —           |
> > | LDF              | 66.17      | 65.60     | 55.28        | 42.06     | 69.59     | 35.81     | 61.94     | 33.27       | 56.80       |
> > | PAPN             | 69.93      | 67.48     | **60.13**    | 45.39     | 72.81     | 35.98     | 59.94     | 35.13       | 58.75       |
> > | ReSA             | 65.82      | 64.76     | 56.70        | 42.53     | 71.31     | 34.92     | 60.46     | 34.64       | 58.84       |
> > | **CLUE (Ours)**  | **69.62**  | **72.66** | 58.59        | **48.53** | **74.71** | **43.45** | **69.49** | **40.66**   | **63.56**   |
> > | EsViT (Swin-T)   | 70.54      | 59.12     | 55.18        | 43.48     | 73.08     | 31.95     | 58.40     | 27.06       | 53.02       |
> > | LoDisc (ViT-B)   | 73.23      | 69.72     | 62.17        | 45.89     | 72.75     | 41.55     | 67.24     | 41.49       | 68.59       |
> > | **CLUE (ViT-B)** | **77.83**  | **75.67** | **62.76**    | **61.93** | **83.91** | **52.05** | **78.14** | 40.26       | 65.11       |
> >
> >
> > Table 2: Flowers / Pets / Food-101
> > Fine-grained classification accuracy (\%) and retrieval performance (Rank-1 / Rank-5, \%) on Flowers, Pets, and Food-101.
> > | Method   | Flower-cls | Pet-cls | Food-cls  | Flower-R1 | Flower-R5 | Pet-R1 | Pet-R5 | Food-R1   | Food-R5   |
> > | -------- | ---------- | ------- | --------- | --------- | --------- | ------ | ------ | --------- | --------- |
> > | LCR      | 85.24      | 90.73   | 73.34     | 90.26     | 97.15     | 89.78  | 97.77  | 52.62     | 74.70     |
> > | ReSA     | 84.17      | 91.82   | 72.91     | 93.89     | 97.92     | 89.24  | 97.87  | 56.08     | 75.99     |
> > | **CLUE** | 86.28  | 91.63   | 83.37 | 95.51 | 98.06 | 90.68  | 97.87  | 67.14 | 83.89 |
> >
> >
> > Table 3: Caltech-256
> > Fine-grained classification accuracy (\%) and retrieval performance (Rank-1 / Rank-5, \%) on Caltech-256.
> > | Method   | Caltech-cls | Caltech-R1 | Caltech-R5 |
> > | -------- | ----------- | ---------- | ---------- |
> > | LCR      | 74.44       | 75.82      | 87.85      |
> > | ReSA     | 84.07       | 78.36      | 88.04      |
> > | **CLUE** | 86.39   | 80.22  | 90.90  |

---

### Meta-Review · Area_Chair_vXwM · 2026-01-11

**Summary:**

This paper studies the limitations of standard self-supervised learning (SSL) methods for fine-grained visual recognition (FGVR), identifying two key failure modes: over-dispersion and over-collapse. To address these, the authors propose CLUE, a multi-level regularization framework combining (i) a global soft-InfoNCE loss, (ii) part-level clustering via VLAD-style local descriptors, and (iii) instance-level semantic guidance from vision–language models (VLMs). The paper further introduces granularity-aware clustering diagnostics and provides empirical results across several fine-grained datasets, with additional evaluations added during rebuttal. Reviewer opinions are highly mixed. While some reviewers appreciate the problem diagnosis and empirical improvements, major concerns remain regarding novelty, positioning relative to prior work, experimental protocol, and reliance on existing components.

**Reviewer Concerns:**

**Concerns addressed by the rebuttal**
- Experimental protocol clarity: The authors clarified the fine-grained SSL setting (self-supervised training on target datasets with ImageNet-initialized backbones), aligning it with prior FGVR SSL works.
- Broader empirical evaluation: Additional datasets (Flowers, Pets, Food-101) and a generic benchmark (Caltech-256) were added; transformer backbones (ViT-B) were also included.
- Ablations on VLM guidance: New ablations show that VLM/text guidance alone is insufficient and works best when combined with vision-only SSL components.
- Computational overhead of VLMs: Clarified as a one-time, offline preprocessing cost with no impact on SSL training/inference.
- Positioning vs. ReSA and hierarchical SSL: The authors clarified that soft-InfoNCE is reused from ReSA and repositioned novelty around a multi-granularity clustering perspective with coupled regularization and diagnostics.

**Outstanding concerns**
- Novelty remains contested: Multiple reviewers (ZrGC, 6fZG, pyDz) remain unconvinced that CLUE introduces sufficiently new ideas beyond existing work on soft-InfoNCE (ReSA), global-local SSL, hierarchical/part-based SSL, and CLIP-guided alignment. The contribution is widely perceived as a composition and refinement rather than a fundamentally new method.
- Dependence on existing components: The framework relies heavily on known ingredients (ReSA-style soft targets, VLAD-like descriptors, VLM embeddings), raising questions about whether the gains stem from integration rather than conceptual novelty.
- Experimental protocol deviates from mainstream SSL: Despite clarification, some reviewers remain concerned that pretraining directly on target FGVR datasets (even with self-supervision) limits comparability to standard SSL pipelines.
- Hyperparameter sensitivity and practicality: The number of part clusters is manually chosen and dataset-dependent, with no adaptive mechanism proposed.
- Evaluation still seen as selective: While expanded, some reviewers remain unconvinced that results demonstrate strong generalization beyond FGVR or that gains over recent baselines are substantial enough to justify acceptance.

**Reviewer Scores:**

- Reviewer ZrGC: 4 (confidence 3) — stated the initial rating is maintained
- Reviewer pyDz: 2 (confidence 3) — reject, no score change indicated
- Reviewer 6fZG: 4 (confidence 5) — borderline, no score change indicated
- Reviewer Npuk: 6 (confidence 3) — accept threshold, no score change indicated

---

### Decision · Program_Chairs · 2026-01-26

Reject